# Production and characterization of the exopolysaccharide from strain *Paenibacillus polymyxa* 2020

Elena V. Liyaskina[1]*, Nadezhda A. Rakova[1], Alevtina A. Kitykina[1], Valentina V. Rusyaeva[1], Philip V. Toukach[2], Alexey Fomenkov[3]*, Saulius Vainauskas[3], Richard J. Roberts[3], Victor V. Revin[1]*

**1** Department of Biotechnology, Bioengineering and Biochemistry of the National Research Mordovia State University, Saransk, Russia, **2** N.D. Zelinsky Institute of Organic Chemistry, Russian Academy of Sciences, Moscow, Russia, **3** New England Biolabs Inc., Ipswich, MA, United States of America

* liyaskina@yandex.ru (EVL); fomenkov@neb.com (AF); revinvv2010@yandex.ru (VVR)

## Abstract

*Paenibacillus* spp. exopolysaccharides (EPSs) have become a growing interest recently as a source of biomaterials. In this study, we characterized *Paenibacillus polymyxa* 2020 strain, which produces a large quantity of EPS (up to 68 g/L), and was isolated from wasp honeycombs. Here we report its complete genome sequence and full methylome analysis detected by Pacific Biosciences SMRT sequencing. Moreover, bioinformatic analysis identified a putative levan synthetic operon. *Sac*C and *sac*B genes have been cloned and their products identified as glycoside hydrolase and levansucrase respectively. The Fourier transform infrared (FT-IR) and nuclear magnetic resonance (NMR) spectra demonstrated that the EPS is a linear β-(2→6)-linked fructan (levan). The structure and properties of levan polymer produced from sucrose and molasses were analyzed by FT-IR, NMR, scanning electron microscopy (SEM), high performance size exclusion chromatography (HPSEC), thermogravimetric analysis (TGA), cytotoxicity tests and showed low toxicity and high biocompatibility. Thus, *P. polymyxa* 2020 could be an exceptional cost-effective source for the industrial production of levan-type EPSs and to obtain functional biomaterials based on it for a broad range of applications, including bioengineering.

## Introduction

Interest in the biosynthesis and production of different EPSs has increased considerably in recent years, because they are potential candidates and precursors for many commercial applications in different industries, including but not limited to food, petroleum, pharmaceuticals and biomedicine [1]. Some of the most used EPSs are xanthan from *Xanthomonas campestris*, dextran produced by *Leuconostoc mesenteroides*, *Streptococcus* sp., *Lactobacillus* sp., alginate from *Azotobacter* sp., *Pseudomonas* sp., gellan derived from *Pseudomonas elodea* and *Sphingomonas paucimobilis*, curdlan from *Alcaligenes faecalis*, hyaluronan from *Streptococcus equi*, pullulan from *Aureobasidium pullulans*, bacterial cellulose from *Komagataeibacter* sp., and

the accession numbers: CP049783-CP049784. The original sequence reads have been deposited at NCBI under SRA SRR11271650 and SRR11271651. Biosample: SAMN14247784. Bioproject:PRJNA609344. The complete genome sequence of P. polymyxa 2020 is available in GenBank under the accession numbers: CP049598-CP049599. The original sequence reads have been deposited at NCBI under SRA: SRR11236808; SRR11236809; SRR11236810; SRR11236811. Biosample: SAMN14247689. Bioproject: PRJNA609344

**Funding:** This work was supported by the Ministry of Science and Higher Education of the Russian Federation, grant number FZRS-2020-0003. New England Biolabs provided support in the form of salaries for authors A.F, S.V. and R.J.R. The specific roles of these authors are articulated in the 'author contributions' section. The funders had no role in study design, data collection and analysis, decision to publish, or preparation of the manuscript.

levan from *Bacillus* sp., *Paenibacillus* sp., *Halomonas* sp. etc. The importance attached to the EPS market and commercial applications has encouraged the rapid search for new producers and the isolation, production and characterization of EPSs in order to obtain novel functional materials with a wide range of applications. This includes the β-(2→6) fructose polymer levan, which is considered to be one of the most promising microbial EPSs for a wide range of industrial applications [2,3]. It has received much attention due to its anticancer, anti-oxidant, anti-bacterial, anti-inflammation, immune-modulating and probiotic activities as well as many outstanding physicochemical properties such as low intrinsic viscosity, high adhesive strength, high water solubility, film-forming ability and high biocompatibility [2–4].

Levan has many potential applications in the food, cosmetic, pharmaceutical, chemical and medical industries [1–4]. In the food industry, levan can be used as a gelling agent as well as a food supplement with probiotic properties [5] such as cholesterol reducing ability. Also, levan stimulates the growth of the intestinal microbiome (*Faecalibacterium*, *Bifidobacterium*, *Lactobacilli*) [3,6]. Moreover, microbial levan can be a good source for pure fructose production. Levan has many possible uses in medicine [3,7]. Indeed, some sulfated, phosphorylated and acetylated derivatives of levan have demonstrated immuno-stimulatory properties and can be used as antitumor and anti-HIV supplements [8,9]. In particular, polysaccharide produced by the halophilic bacterium *Halomonas smyrnensis* has been shown to have immuno-stimulatory properties [10,11]. Levan can also be used as a surfactant agent during targeted drug delivery with a prolonged mode of action [3,12]. Additionally, it can provide biocompatible surfaces [13,14], especially in tissue engineering [7,15,16]. Levan has potential film forming and high adsorption properties [3]. Therefore, it can be used as an adsorbent for heavy metals in industrial waste water treatment. Levan has a higher adhesive property than carboxymethylcellulose and has shown utility for gluing wood in a biocomposite material design [17].

Many species of bacteria can synthesize levan, including acetic acid producing *Gluconobacter oxydans*, *Acetobacter aceti*, some species of *Pseudomonas*, *Erwinia* (*herbicola* group), *Aeromonas*, *Bacillus*, *Zymomonas*, *Corynebacterium*, *Lactobacillus*, *Aerobacter*, *Steptococcus*, *Azotobacter* and so on [2,3]. Currently more than 100 species of bacteria have been shown to produce levan [3]. Gram-positive bacteria are the most studied producers of levan, including *Bacillus subtilis*. It was shown that upon cultivation for 21 hours in media supplemented with 20% sucrose, these bacteria can produce up to 40–50 g/L of levan [18]. During batch and continuous cultivation in a bioreactor the yield of levan was 61 and 100 g/L respectively [19]. Moreover the *B. subtilis* strain CCT 7712 was able to synthesize up to 111.6 g/L of levan from 400 g/L sucrose media during 16 h cultivation [20].

The possible producers of levan were also identified among gram-negative halophilic bacteria of the *Halomonas* genus [2,21], including *Halomonas smyrnensis* AAD6T [22]. Their ability to grow in high concentrations of NaCl can be used to resolve the problem of sterility in an industrial setting. *Zymomonas mobilis* also produces levan as originally reported in the 1960s [23]. The original strains ZAG-12 [24] and ATCC 31821 [25] were able to produce levan with approximate yields of 14.67 and 21.69 g/L respectively. It was reported that during continuous cultivation of *Z. mobilis* CCT4494, when immobilized in Ca- alginate gel the amounts of levan can range from 18.84 up to 112.53 g/L dependent on the incubation time [26]. *Komagataeibacter xylinus*, the producer of bacterial cellulose can also produce levan [27,28]. Several recombinant *E.coli* strains have been developed to study the biochemistry of levan synthesis by cloning and expression of levansucrase genes from *Leuconostoc mesenteroides* and *Bacillus amyloliquefaciens* [29,30].

*Paenibacillus* spp. EPSs have recently attracted a great deal of interest, because of their biotechnological potential in different industrial and wastewater treatment processes [31,32]. *Paenibacillus* spp. can produce a wide variety of different EPSs with diverse structures, like levan

[32,33] and curdlan [34]. *Paenibacillus polymyxa* is a typical member of the *Paenibacillus* genus, that include about 200 different species of gram-positive spore-forming bacteria [31,35]. *P. polymyxa* strain ATCC 842[T] (= DSM 36[T] = KCTC 3858[T]) has been deposited in a number of microbial collections. This bacterium was very often associated with the plant root microbiome, where it participates in bioprotection by the synthesis of antibiotics, phytohormones, hydrolytic enzymes and EPS [36–39]. Despite the importance of these bacteria, genome information is still limited. Full genome sequencing and assembly has been done for only few strains of *Paenibacillus* including *P. polymyxa* E681 [40], *P. polymyxa* SC2 [41], *P. polymyxa* ATCC 842[T] [42], *P. polymyxa* KF-1, soil isolated producer of antibiotics [43] and *Paenibacillus sp.*Y412MC10 [44], isolated from Obsidian Hot Spring in Yellowstone National Park, Montana, USA. Most of the deposited genome sequences are still in shotgun form.

The yield, structure and physical-chemical properties of *P. polymyxa* EPSs depend on many factors such as cultivation condition, source of carbon, C/N ratio, pH of media and time of cultivation [38,45] Levan biosynthesis occurs extracellularly from sucrose under the influence of the enzyme levansucrase (EC 2.4.1.10) [2,46,47]. The enzyme breaks down sucrose into glucose and fructose, with the remainder of the latter transferred to the sucrose acceptor molecule [48]. It has been reported that levansucrase, with strong sucrose hydrolysing activity, is involved in many *P. polymyxa* strains and may be responsible for the high yield of EPS with sucrose as a carbon source [33,38]. Thus, the strain *P. polymyxa* 92, isolated from the roots of wheat forms 38.4 g/L of levan in 10% sucrose media [33]. Sugar-containing waste from food production, such as molasses, was used to reduce the cost of levan production [49]. It has been reported that levan was obtained on molasses media with the help of bacteria such as *Azotobacter vinelandii* VKPM B-5787 [17], *Bacillus lentus* V8 [50], *Microbacterium laevaniformans* PTCC 1406 [51], *Paenibacillus polymyxa* NRRL B-18475 [52] and *Zymomonas mobilis* ATCC 31821 [25]. The release of the polysaccharide, formed by *Bacillus licheniformis* NS032, in a molasses-containing medium with an estimated sucrose concentration of 62.6% lead to 53.2 g/L levan yield [49].

The aim of this work was to characterize the properties of *Paenibacillus polymyxa* 2020, a new producer of levan polysaccharide, its full-genome sequence, methylome analysis and bioinformatic analysis of the levan producing operon. The structure and properties of the levan polymer produced from sucrose and molasses were analyzed by FT-IR, NMR, SEM, HPSEC, TGA and cytotoxicity tests to obtain functional biomaterials for a wide range of applications, including biomedicine.

## Material and methods

### Bacterial strain and culture conditions

The strain *Paenibacillus polymyxa* 2020 was isolated from wasp honeycombs in 2019 at the Department of Biotechnology, Bioengineering and Biochemistry of the National Research Mordovia State University, Saransk, Russia. The strain was deposited in the Russian National Collection of Microorganisms (VKM) (Accession No. VKM: B-3504D). *P. polymyxa* 2020 was grown and maintained on a sucrose medium consisting of (g L$^{-1}$): sucrose 140, CaCO$_3$ 0.02, NaCl 0.01, K$_2$HPO$_4$ 0.01, MgSO$_4$×7H$_2$O 0.02; MnSO$_4$ 0.01; FeSO$_4$ 0.01, and agar 20.0. pH 7.2. For EPS production, the following media were used: sucrose medium consisting of (g L$^{-1}$): sucrose, yeast extract 7, K$_2$HPO$_4$ 2.5, NH$_4$SO$_4$ 1.6, MgCl$_2$ 0.4; molasses medium with a sucrose content of (g L$^{-1}$) 50, 100, 150 and 200. The initial pH of the media was 7.2. The effect of sucrose on the yield of EPS was analyzed with 50, 100, 150, and 200 g L$^{-1}$ of sucrose. The sugar beet molasses used in this study was obtained from the Romodanovsky Sugar Factory, Russia. The beet molasses contained total solid 82.8%, sucrose 50.0%, glucose 0.4%, total nitrogen

2.4%, and total phosphorus 0.02%. The culture media were autoclaved for 20 min at 120˚C. The medium was inoculated with 10% (v/v) inoculum. To prepare the inoculum, *P. polymyxa* 2020 was transferred aseptically from an agar plate into a 250 mL Erlenmeyer flask containing 100 mL of culture medium and incubated on a shaker incubator (Model ES-20/60, Biosan, Latvia) at 30˚C, and 250 rpm for 24 h. EPS was produced in 500 mL Erlenmeyer flasks containing 100 mL of culture medium on a shaker incubator at 30˚C, and 250 rpm for 96 h. The experiments were conducted in triplicate, and the mean is reported.

## Genome sequencing and analysis of *P. polymyxa* 2020 and *P. polymyxa* DSM36

SMRTbell libraries were prepared using a modified PacBio protocol adapted for NEB reagents. Genomic DNA samples were sheared to an average size of ~ 6–10 kb using the G-tube protocol (Covaris; Woburn, MA, USA), treated with FFPE, end repaired, and ligated with hairpin adapters. Incompletely formed SMRTbell templates were removed by digestion with a combination of exonuclease III and exonuclease VII (New England Biolabs; Ipswich, MA, USA). The qualification and quantification of the SMRTbell libraries were made on a Qubit fluorimeter (Invitrogen, Eugene, OR) and a 2100 Bioanalyzer (Agilent Technologies, Santa Clara, CA). SMRT sequencing was performed using a PacBio RSII (Pacific Biosciences; Menlo Park, CA, USA) based on standard protocols for 6–10 kb SMRTbell library inserts. Sequencing reads were collected and processed using the SMRT Analysis pipeline from Pacific Biosciences (http://www.pacbiodevnet.com/SMRT-Analysis/Software/SMRT-Pipe) [53]. Next-generation SMRT sequencing technology from Pacific Biosciences Inc. allowed the assembly of a complete circular genome. It also enabled the determination of the m6A and m4C modified motifs. [54–56].

## EPS isolation and purification

The culture broth was diluted 1:2 with deionized water to decrease viscosity and was centrifuged at 12000×g for 30 min to separate cells from the supernatant liquid. The pellet obtained was washed with distilled water to remove adsorbed polysaccharide and dried to a constant weigh at 105˚C. The growth of *P. polymyxa* 2020 was determined by measuring dry cell weight.

After removal of biomass by centrifugation, EPS was obtained from the supernatant by two successive precipitations using two volumes of ice-cold ethanol. The precipitate was dissolved and dialyzed (molecular weight cut-off 14000, Sigma D9402) against running water for at least three days to completely remove low molecular weight impurities. Repeated precipitation and dissolution in water was performed to purify the EPS. The crude EPS was dried to a constant weight at 90˚C. EPS production has been reported as gram dry weight of EPS per litre of the medium (g $L^{-1}$).

## HPLC determination of sugars

The culture broth was centrifuged on a Sorvall RC-6 Plus high-speed centrifuge (USA) at 12000 ×g for 30 min to separate the biomass. The resulting supernatant was mixed with acetonitrile 1:3 (v/v) and centrifuged in an ELMI CM-50 laboratory centrifuge (Latvia) at 10000 ×g for 5 min. The qualitative detection of sugars was carried out by high-performance liquid chromatography (HPLC) using an LC-20A high-performance liquid chromatograph with a refractometric and UV detector (Shimadzu, Japan). The determination of sugars was carried out by UV detection at 254 nm using a Supelcosil LC-NH$_2$ column (250x4.6 mm, 5 μm, USA). Chromatographic analysis was performed in a gradient mode with an eluent flow rate of 0.4 ml/min

and a column thermostat temperature of 40˚C. The mobile phase was composed of 75% acetonitrile.

## FTIR spectra

The crude EPS was freeze-dried using a FreeZone Plus freeze-dryer lyophilizer (Labconco, USA) and crushed into powder form, mixed with potassium bromide, and pressed into a small tablet that was subjected to Fourier transform infrared (FTIR) spectroscopy using an IRPrestige-21 Fourier transform infrared spectrometer (Shimadzu, Japan) in absorption mode. For each sample, 32 scans were collected at a resolution 4 $cm^{-1}$ with wave numbers ranging from 4000–400 $cm^{-1}$.

## NMR spectroscopy

$^{1}$H and $^{13}$C NMR spectra of the hydrolyzed EPS were recorded using a JEOL JNM-ECX400 spectrometer (JEOL, Japan) (400 and 100 MHz respectively) in a $D_2O/H_2O$ solution (60 mg of substance in 0.7 ml $D_2O$ and 0.005 ml acetonitrile) with and without 70% $HClO_4$. The signals ($\delta H$ 2.06 ppm, $\delta C$ 119.68 and 1.47 ppm) of acetonitrile were used as a chemical shift reference. The spectra were processed using the software packages ACD/NMR Processor Academic Edition, ver. 12.01 and Delta 4.3.6. The NMR spectra of the native polysaccharide were recorded using a Bruker AV600 NMR instrument (600 MHz) at 40˚C with HDO signal suppression. To prepare the NMR sample, 20 mg of the polysaccharide were lyophilized and dissolved in 99.9% $D_2O$. The NMR spectra were processed by MestreLabs Mestre Nova software and referenced to internal DSP at $\delta_H$ 0.00, $\delta_C$ -1.59.

## Scanning electron microscopy (SEM)/Energy-dispersive X-ray spectroscopy (EDX)

The topography and morphology of the EPS surface were examined with a Quanta 200 I 3D FEI scanning electron microscope (USA). The EPS solution at a concentration of 10% (w/v) was frozen in a low-temperature cooler MDF-U53V (SANYO, Japan) at −50˚C for 24 h, and further dried on a FreeZone Plus freeze-dryer lyophilizer (Labconco, USA). An analysis of the sample composition was performed using energy-dispersive X-ray spectroscopy (EDX).

## Measurement of molecular weight

The molecular weight distributions of the exopolysaccharides (100 mg/mL, injection volume: 20 μL) were determined using HPSEC with refractive index (RI) detection. Analysis was performed on an LC-20A system (LC-20AD pump, RID-10A detector, Shimadzu, Japan) equipped with an Ultrahydrogel Linear column (300mm×7.8mm i.d., 10 μm particle size, «Waters», USA) and an Ultrahydrogel Guard Columns (40mm×6mm i.d., 6 μm particle size, Waters», USA). Water at 0.4 mL/min was used as eluent at 40˚C, because all analyzed samples were readily soluble in water. Dextran standard compounds with known molecular weights (670 kDa, 250 kDa, 80 kDa, 25 kDa, and 12 kDa, Sigma Aldrich, Denmark) were used to estimate the molecular weight of the exopolysaccharide fractions.

## Thermogravimetric analysis (TGA)

Thermogravimetric analysis was performed using a TG 209 F1 Libra thermobalance (Netzsch, Germany).

## Cytotoxicity tests

The L929 mouse fibroblast cell culture used in this study was obtained from the tissue culture collection of the D.I. Ivanovsky Institute of Virology, Russia. The cells were cultured in Dulbecco's modified Eagle's medium (DMEM) (Paneko, Russia) containing 10% fetal bovine serum (FBS) (HyClone, USA) in the presence of penicillin and streptomycin under standard conditions: 5% $CO_2$ atmosphere, t = 37˚C, 5% humidity in an MCO-170M incubator (Sanyo, Japan). Cells in exponential growth phase were dispensed into a 96-well plate ($5 \times 10^3$ cells/well). After 24 hr., fresh medium, containing the EPS was added to the 96 wells and incubated for a further 24hr. Wells without EPS were used as a control. The morphological structure of the cells were monitored using an inverted optical microscope (Micromed, Russia). The MTT assay was used to measure cytotoxicity of the EPS. The medium was replaced with a fresh one containing 5 mg/mL dimethyl thiazolyl diphenyl (MTT). After 4 h incubation time the medium was removed and 150 µL DMSO was added. The optical density was measured on a microplate reader EFOS 9305 (Russia) at a wavelength of 492 nm with a reference wavelength of 620 nm. Cell viability was recorded as the ratio of the optical density of the sample to the control and expressed as a percentage.

## Cloning and expression of *sac*B and *sac*C genes

Bacterial *Escherichia coli* strains, plasmids and oligonucleotides are listed in **S1 Table.** Cultures were grown in Luria-Bertani (LB) broth or agar on appropriate antibiotics. All restriction enzymes, DNA and protein markers were from NEB, MA. Q5 "Hot Start" DNA polymerase (M0543, NEB) was used for PCR amplification of genes for cloning.

PCR fragments for *sac*B (ORF502), *sac*C (ORF530) and sacBC (ORF502-503) have been amplified with primers P1-P2, P3-P4 and P1-P4 respectively and cloned into an NdeI-BamHI linear pSAPv6 vector under the control of a T7 promoter by modified Gibson assembly technology with HiFi Builder gene assembly kit (E5520S NEB, MA). After transformation into the *E.coli* strain ER2683, the resulting clones were screened by colony PCR with S1271-S1248 T7 primers and restriction digestion. The resulting pPpo2020_SacB, pPpo2020_SacC and pPpo2020_SacBC expression plasmid DNA constructs were sequenced on the ABI DNA sequencer with a set of S1248 and S1271 sequencing primers. The sequences were assembled using Laser Gene and analyzed using BLAST (http://www.ncbi.nlm.nih.gov/). pPpo2020_-SacB, pPpo2020_SacC and pPpo2020_SacBC plasmid DNAs were used as templates for quick protein expression and enzymatic assay in the PURExpress *in vitro* transcription-translation system (E6800S, NEB, MA).

## SacB and Sac C activity assay and oligosaccharide analysis

To test the hydrolysis/transfer activity of putative SacB, SacC proteins, 1 µl of *in vitro* expressed protein was incubated with 50 mM sucrose, 10 mM MES buffer, pH 6.0 in 10 µl total reaction volume overnight at 37˚C. 2.5 µl of the reaction mixture was added to 17.5 µl of acetonitrile and the sample was analyzed by UPLC-HILIC-MS. Samples were separated by UPLC using an ACQUITY UPLC glycan BEH amide column (2.1 x 150 mm, 1.7 µm) from Waters on an H-Class ACQUITY instrument (Waters). Solvent A was 50 mM $NH_4F$ buffer pH 4.4 and solvent B was acetonitrile (ACN). The gradient used was 0–35 min, 12–47% solvent A; 35.5–36 min, 70% solvent A; 36.5–40 min, 12% solvent A. The flow rate was 0.4 ml/min. The injection volume was 18 µl and the sample was prepared in 88% (v/v) ACN. Samples were kept at 5˚C prior to injection and the separation temperature was 40˚C. Conditions for inline mass detection using the ACQUITY quadrupole QDa (Waters) were as follows: Electrospray ionization (ESI) in negative mode; capillary voltage, 0.8 kV; cone voltage, 25 V; sampling frequency, 5

Hz; probe temperature 400˚C. The QDa analysis worked using full scan mode (MS Scan, the mass range was set at m/z 100–1250), and selected ion recording (SIR, the mass was set at m/z 179 and/or 341). A Waters Empower 3 chromatography workstation software was used for data processing including a traditional integration algorithm, no smoothing of the spectra and manual peak picking.

## Results and discussion

Members of the *Paenibacillus* genus have been isolated from different environments. While many of the species are relevant to humans, animals, and plants, the majority are found in soil, often associated with plant roots to promote growth They can be to improve agriculture [31].

### Complete genome sequence, assembly and methylome analysis

A 6 kb SMRTbell library was sequenced using C4-P6 chemistry and run on 4 SMRT cells with a 360-minute collection time. 1.6 Gb of data were collected from 55,864 sequencing reads with 1,705 bp mean subread lengths. This sequence was assembled *de novo* using the HGAP_Assembly.3 version 2.3.0 with default quality and read length parameters and polished once using Quiver. The original polished assembly generated 125 linear contigs (the sum of the contigs equaled 5,907,830 bp) with N50 equal to 104,550bp. BLAST analysis at NCBI of the 125 contig assembly of *Paenibacillus polymyxa* 2020 identified the strongest hits as having 99.5% homology to AFOX01 a shotgun Miseq-based sequence assembly of *P. polymyxa* DSM36/ATCC842 [42].

*P. polymyxa* DSM36 native genomic DNA (2 μg) was obtained directly from the DSMZ collection and a 15kb SMRT bell library was prepared according to the Pacific Biosciences 10–20 kb protocol including additional separation using a Blue Pippin and sequenced with C4-P6 chemistry on 2 SMRT cells. One library consisted of non-size selected (15 kb) DNA and one contained size-selected (17kb) DNA and a 360-minute collection time was used for each sample on the RSII PacBio sequencer. 28,951 sequencing reads with 12,000 bp mean subread lengths, yielded 0.35 Gb of data. This was *de novo* assembled using the HGAP_Assembly.3 version 2.3.0 with default quality and read length parameters and polished 3 times using Quiver. The polished assembly generated 2 closed circular genomic elements of 5,919,686 bp with 45.08% GC content for the main chromosome and 45,518 bp with 41.76% GC for the plasmid pPpo45. The assembled sequences were submitted to the NCBI Prokaryotic Genomes Annotation Pipeline (PGAP) [57,58].

A re-run of the original reads from *P. polymyxa* 2020 against the genomic reference sequence of *P. polymyxa* DSM36 and polishing 5 times using Quiver demonstrated that 97.7% of the *P. polymyxa* 2020 reads mapped to the *P. polymyxa* DSM36 reference and generated a final genome with 99.99% homology between the two strains. The same m6A modified DNA motif, CN**A**GNNNNNTT**G**K, was detected in both *P. polymyxa* 2020 and DSM36 strains and can be assigned to a Type I restriction-modification system Ppo36I in the DSM36 strain and Ppo2020I in the 2020 strain (**Fig 1**), consistent with the notion that these two *P. polymyxa* strains are essentially identical. All data have been deposited at NCBI and in REBASE [59].

**Nucleotide sequence accession numbers.** The complete genome sequence of *P. polymyxa* DSM36 was deposited in NCBI and given the accession numbers: CP049783-CP049784. The original sequence reads also have been deposited at NCBI under SRA SRR11271650 and SRR11271651.Biosample: SAMN14247784. The complete genome sequence of *P. polymyxa* 2020 also is available in GenBank with the accession numbers: CP049598-CP049599. The original sequence reads have been deposited at NCBI under SRA: SRR11236808; SRR11236809; SRR11236810; SRR11236811. Biosample: SAMN14247689. Data for both strains are available under Bioproject: PRJNA609344.

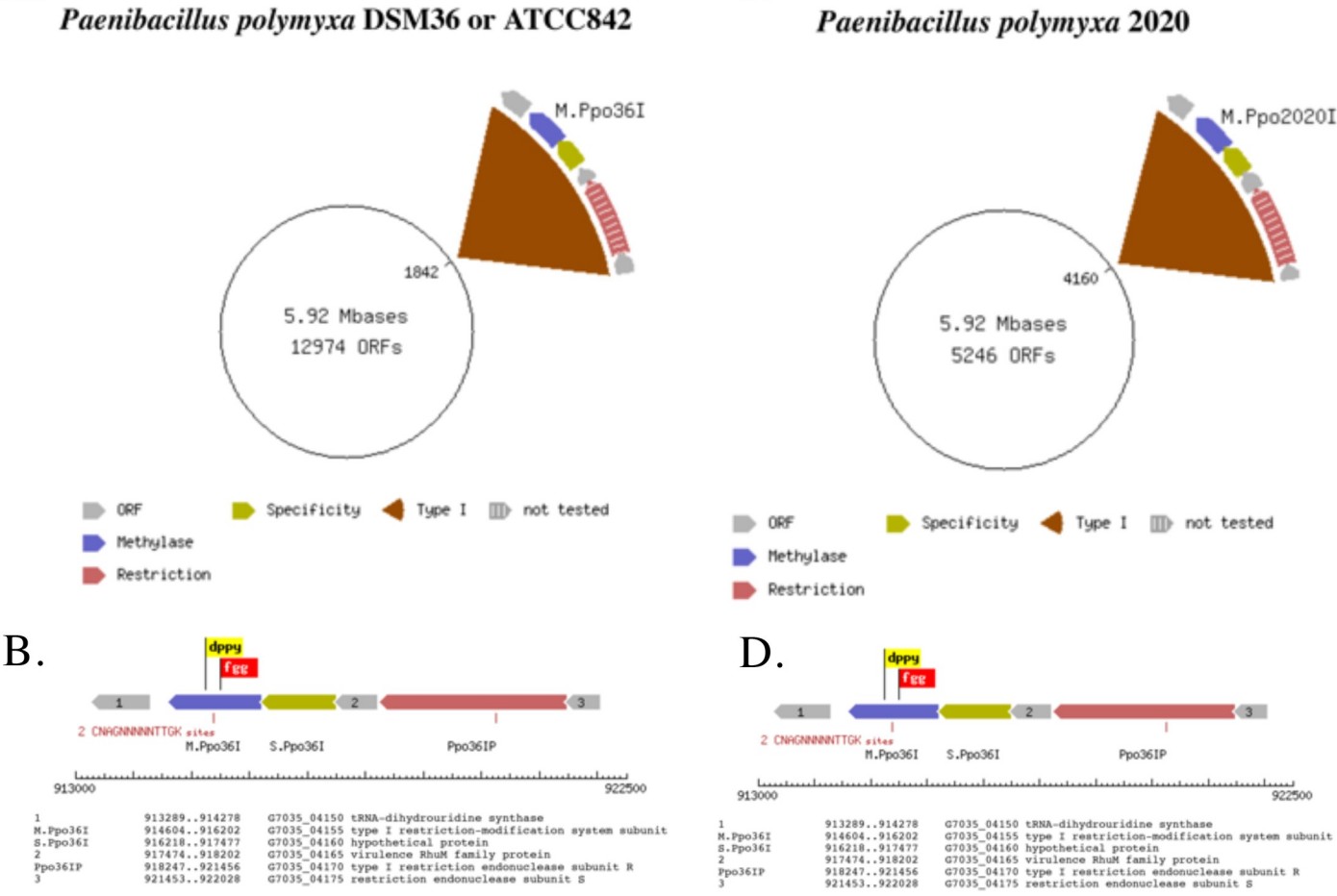

**Fig 1.** Chromosome maps and Type I restriction-modification systems identified in *P. polymyxa* DSM36 (A and B) and *P. polymixa* 2020 (C and D) indicates they are identical species.

The genome sequences of 212 strains of *Paenibacillus*, that represent 82 different species are available [31]. Genome sizes range from 3.02 Mb for *P. darwinianus* Br, isolated from Antarctic soil [60] to 8.82 Mb for *P. mucilaginosus* K02, implicated in silicate mineral weathering [61] and the number of genes varies from 3,064 for *P. darwinianus* Br to 8,478 *P. sophorae* S27. The insect pathogens *P. darwinianus*, *P. larvae* and *P. popilliae* have smaller genomes from 4.51 and 3.83 Mb, respectively, perhaps reflecting their niche specialization. The GC content of *Paenibacillus* DNA ranges from 39 to 59% [62].

## EPS synthesis by *P. polymyxa* 2020

In this work, we studied the formation of EPS in a sucrose and molasses media containing sucrose at concentrations of 50, 100, 150 and 200 g $L^{-1}$. **Fig 2A** shows the kinetics of levan formation by *P. polymyxa* 2020 grown in a medium with sucrose. In medium with an initial 50 g $L^{-1}$ sucrose, maximum EPS concentration was reached at 48 h (24.38 g $L^{-1}$) and this concentration remained relatively constant until the end of the experiment (96 h). The highest yield of EPS was 53.78 g $L^{-1}$ with an 150 g $L^{-1}$ sucrose at 96 h. In medium with an initial 100 g $L^{-1}$ and 200 g $L^{-1}$ of sucrose, the maximum production of EPS was reached at 96 h (42.02 g $L^{-1}$ and

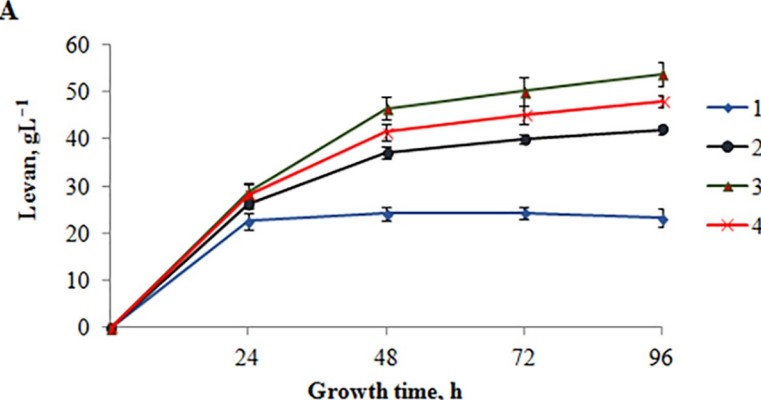

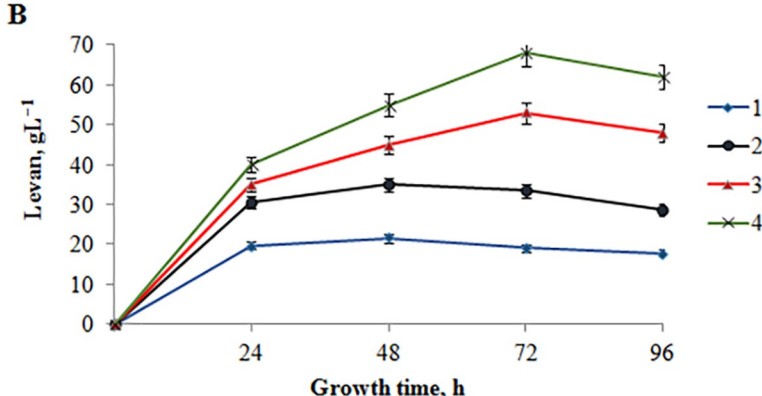

**Fig 2.** EPS accumulation during growth of *P. polymyxa* 2020 in a sucrose (A) and molasses (B) media containing sucrose at concentrations 50 g L$^{-1}$ (1), 100 g L$^{-1}$ (2), 150 g L$^{-1}$ (3), 200 g L$^{-1}$ (4).

47.91 g L$^{-1}$ respectively). A similar trend in EPS production was described by Liu et al. [38] for the strain *P. polymyxa* EJS-3 and by Grinev et al. [33] for the strain *P. polymyxa* 92. However, *P. polymyxa* 2020 produced larger amounts of EPS at the same substrate concentration in the culture medium, as compared with strain *P. polymyxa* EJS-3 and *P. polymyxa* 92. With 15% sucrose, strain *P. polymyxa* 2020 produced the largest amounts of EPS (53.78 g L$^{-1}$), whereas *P. polymyxa* EJS-3 strain produced much less EPS (22.82 g L$^{-1}$) with 16% sucrose [38]. With 10% sucrose, strain *P. polymyxa* 2020 produced 42.02 g L$^{-1}$ of EPS, whereas strain *P. polymyxa* 92 produced less EPS (38.4 g L$^{-1}$) [33]. *P. polymyxa* 92 was grown in 0.5 L Erlenmeyer flasks, containing 150 mL of the medium with initial pH value of 7.2, at 30°C for 3–7 days in an orbital shaker–incubator (180 rpm).

The use of sugar-based raw materials as nutritive substances can greatly help to reduce the production cost of microbial polysaccharides [49]. Molasses, a food industry by-product of cane or beet sugar production, contains up to 80% of dry matter, of which sucrose represents ~48%. Molasses also contains amino acids, organic acids and their salts, betaine, mineral compounds, and vitamins. The kinetics of levan accumulation by *P. polymyxa* 2020 grown in a molasses medium with sucrose in the initial concentration of 50 g L$^{-1}$, 100 g L$^{-1}$, 150 g L$^{-1}$ and 200 g L$^{-1}$ is shown in **Fig 2B**.

In medium with an initial 50 and 100 g L$^{-1}$ sucrose, maximum EPS concentration was reached at 48 h (22.0 g L$^{-1}$ and 35.0 g L$^{-1}$ respectively). The highest yield of EPS was 68.0 g L$^{-1}$ with 200 g L$^{-1}$ sucrose at 72 h. It was higher than in sucrose medium. In medium with an initial

150 g L$^{-1}$ of sucrose, the maximum production of EPS also was reached at 72 h (53.0 g L$^{-1}$). In comparison to the media with sucrose, in the molasses medium, the maximum production of polysaccharide was achieved over a shorter time interval (48–72 h). This data is consistent with previous observations where in contrast to high concentration sucrose media with 200 and 400 g L$^{-1}$, *Bacillus licheniformis* NS032 grows faster on the molasses optimized medium, therefore the maximum production of polysaccharide (53.2 g L$^{-1}$) was achieved over a shorter time interval [49]. Fermentations were conducted in 500 mL flasks with 200 mL of medium with initial pH value of 7.2 in an incubator at 37°C without agitation for 5 days.

Previously molasses has already been used for levan production by *P. polymyxa* [52]. For example, *P. polymyxa* NRRL B- 18475 produced up to 38 g L$^{-1}$ of levan. The high yield of levan was achieved on sugar beet molasses comparable to the yield in sucrose medium (36 g L$^{-1}$) [52]. The fermentation was carried out in the media with an initial pH of 7.0 at 30°C on a rotary shaker at 150 rpm. However, *P. polymyxa* 2020 under similar conditions can produced larger amounts of EPS in the molasses medium, comparable with *B. licheniformis* NS032 and *P. polymyxa* NRRL B- 18475 strains. Growth curves of *P. polymyxa* 2020 in a sucrose and molasses media containing sucrose at concentrations of 50, 100, 150 and 200 g L$^{-1}$ are shown in S1(A) and S2(A) Figs. In sucrose medium maximum biomass was achieved with an initial sucrose 150 and 200 g L$^{-1}$ at 72 h (3.08 g L$^{-1}$) (S1(A) Fig). After that time, a slow decline of biomass was detected. In medium with an initial sucrose concentration of 50 and 100 g L$^{-1}$ the amount of biomass was less and maximum biomass was achieved at 48–72 h (2.0–2.2 g L$^{-1}$). Previous studies also indicated that increased concentrations of sucrose results in higher microbial growth and hence increased levan production [33,49]. In molasses medium maximum biomass was also achieved with an initial sucrose concentration of 150 and 200 g L$^{-1}$ but at an earlier time (48–72 h) (3.31–3.74 L$^{-1}$) (S2(A) Fig).

The pH changes during fermentation on sucrose and on molasses media were compared (S1(B) and S2(B) Figs). In sucrose medium, a rapid drop of pH occurred during the first 24 h due to acid formation, and afterwards, the pH remained stable in the range of 5.0–5.2 (with an initial sucrose concentration of 50 g L$^{-1}$) or 4.3–4.8 (with initial sucrose concentrations of 100, 150 and 200 g L$^{-1}$) until the end of fermentation (S1(B) Fig). The pH, in molasses medium, was only slightly reduced and ranged from 5.79 to 6.0 throughout the entire fermentation (S2(B) Fig). Therefore, pH curves were in accordance with previously reported data [49,75].

To detect the consumption of sucrose during growth of *P. polymyxa* 2020 in a sucrose and molasses media, HPLC analysis of samples was performed (S4 Fig). The bacterial conversion of sucrose into levan leads to glucose accumulation, as shown by the decrease in sucrose levels and the accompanying increase in levan and glucose levels in the growth medium. A small amount of fructose was also detected. The sucrose level dropped and levan started to appear during 24 h fermentation; thereafter, the sucrose level rapidly decreased as levan increased. Levan levels peaked at 4 days and reached about 36% when grown on 15% sucrose medium, constituting a 47% theoretical yield based on the available fructose.

The maximum amount of glucose and fructose in the medium was found at 24 h of *P. polymyxa* 2020 growth. Subsequently, their concentration gradually decreases by 96 h. In the medium with a sucrose concentration of 50 g L$^{-1}$, sugar is practically depleted after 96 h of cultivation. In a medium with a sucrose concentration of 200 g L$^{-1}$, a significant amount of sucrose (22 g L$^{-1}$) and glucose (40 g L$^{-1}$) remains after 96 h of cultivation. As in the media with sucrose, the maximum amount of glucose and fructose in the medium with molasses was found at 24 h of *P. polymyxa* 2020 growth. Subsequently, their concentrations gradually decrease, but by 96 h, a greater amount of glucose remains in the molasses medium than in the sucrose media. In a medium with a sucrose concentration of 50 g L$^{-1}$, 9.87 g L$^{-1}$ glucose remains for 96 h of cultivation. In a medium with a sucrose concentration of 100 g L$^{-1}$, 14.49 g L$^{-1}$

glucose remains for 96 h of cultivation. In a medium with a sucrose concentration of 150 g L$^{-1}$, 22.31 g L$^{-1}$ glucose remains for 96 h of cultivation. In a medium with a sucrose concentration of 200 g L$^{-1}$, 42.11 g L$^{-1}$ glucose remains for 96 h of cultivation. At the same time, sucrose and fructose are practically absent in the medium for 96 h of cultivation.

## Structural characterization of EPS

**FTIR and NMR spectroscopy.** After EPS isolation and purification, several analytical techniques were used to check the polymer structure. The EPS was examined by FTIR and NMR spectroscopy. **Fig 3** shows the FTIR spectra of the levan and EPSs obtained during growth of *P. polymyxa* 2020 in a sucrose and molasses media. The IR spectrum of the control sample of levan was identical to the IR spectrum given in Yu et al. [63]. The EPS samples of *P. polymyxa* 2020 obtained on a nutrient medium with sucrose and molasses were identical to levan, which unequivocally proves that it is levan that is produced by the *P. polymyxa* 2020 strain on a both medium with molasses and with sucrose.

The FT-IR spectrum of levan showed a strong, wide band between 3600 cm$^{-1}$ and 3200 cm$^{-1}$ corresponding to O-H stretching vibrations that exist in polymeric alcohol. Moreover, peaks around 3437, 2928 and 2885 cm$^{-1}$ corresponding to O-H stretching vibrations and C-H stretching and bending vibrations indicated the existence of polysaccharide. The bending vibration of O-H associated with an intense peak around 1651 cm$^{-1}$ also indicated the presence of bound water. The stretching vibrations of C-O-C in pyranose or furanose were attributed to peaks around 1126 and 1014 cm$^{-1}$, while peaks around 922 and 806 cm$^{-1}$ were associated with the symmetrical stretching vibration of furanose and the bending vibration of D-type C-H bonds present in furanose, respectively. Thus, typical signals from furanose were completely different from the signals of pyranose, enabling these two components to be easily distinguished. The existence of C-O-C signals was in accordance with this result. Therefore, preliminary assessment of EPS structure showed that it was composed mostly from a D-furanose with no substituent groups. Thus, the FTIR spectrum demonstrates that the EPS is a fructose polymer (**Fig 3**)

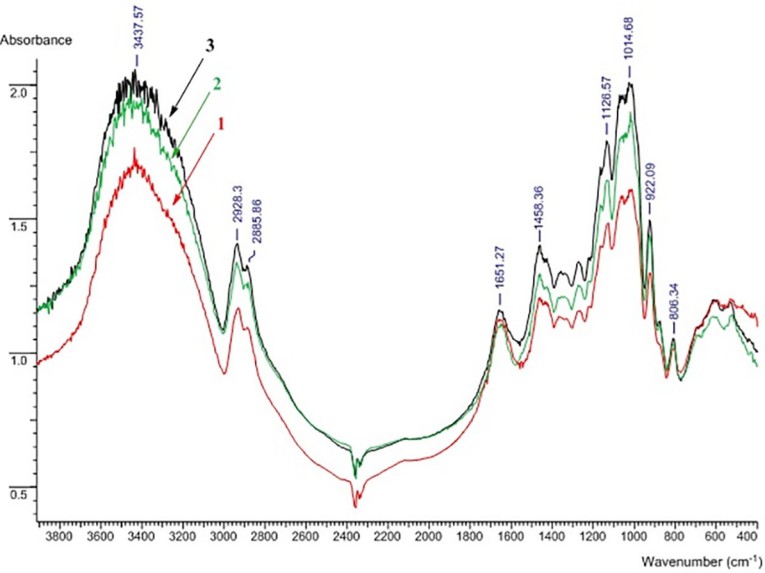

**Fig 3. Overlayed IR spectra of levan (1), EPS of *P. polymyxa* 2020 from sucrose medium (2) and EPS of *P. polymyxa* 2020 from molasses medium (3).**

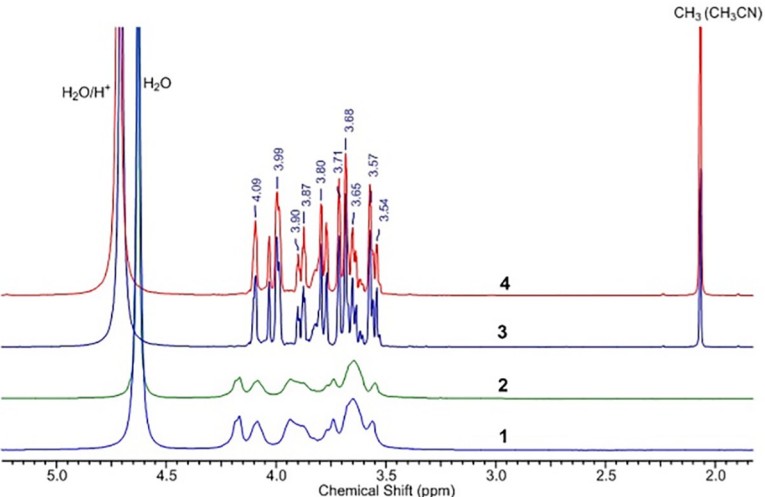

**Fig 4. Overlay of $^{1}$H NMR spectra of the EPS samples of *P. polymyxa* 2020 hydrolyzed with HClO$_4$ from a sucrose medium (1) and fructose (2).**

Due to the high viscosity of water solutions of the native polysaccharide of *P. polymyxa* 2020 we started the structure elucidation from acid-hydrolyzed samples. HClO$_4$ was added to the test solutions: 0.01 ml of acid per 1 ml of the solution, after which the solution was kept at room temperature for 24 h. As a result of acid-catalyzed hydrolysis, a decrease in the solution viscosity was observed. Well–resolved $^{1}$H and $^{13}$C NMR spectra of the hydrolyzed products were obtained. The superimposed $^{1}$H NMR spectra of the EPS samples from *P. polymyxa* 2020 hydrolyzed with HClO$_4$ from a sucrose and fructose media are shown in **Fig 4**.

Fructose in solution has the following distribution of cyclic forms: α-D-fructofuranose— 5.5%; β-D-fructofuranose—22.3%; α-D-fructopyranose—0.5%; β-D-fructopyranose is 71.4% and it can change with temperature and pH of the solution [64]. This led to overlaps of signals from various fructose forms in the $^{1}$H NMR spectrum (**Fig 5**). As shown by the example of cellulose ethers [65], $^{13}$C NMR spectroscopy of polysaccharide hydrolysates is a convenient method for determining their structure. Comparison of the $^{13}$C NMR spectra of the EPS samples of *P. polymyxa* 2020 hydrolyzed with HClO$_4$ and the standard sample of D (-)-fructose

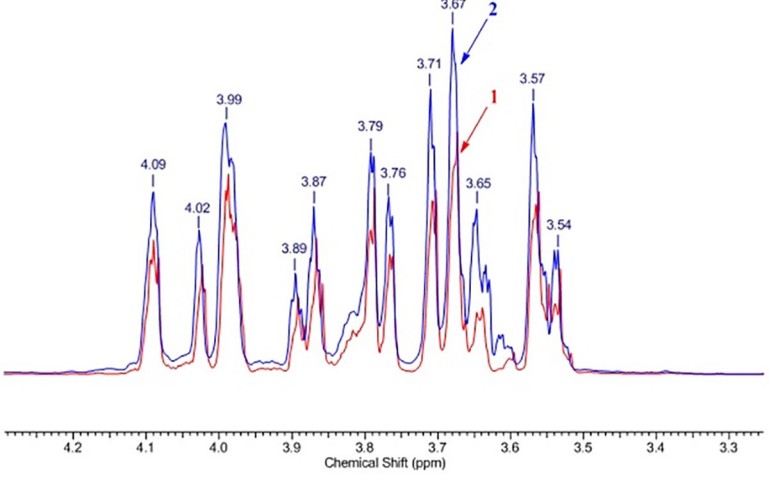

**Fig 5. $^{1}$H NMR spectra of the samples of EPS *P. polymyxa* 2020: Not hydrolyzed: 1 –from molasses medium, 2 – from a sucrose medium; hydrolyzed with HClO$_4$: 3 –from molasses medium, 4—from a sucrose medium.**

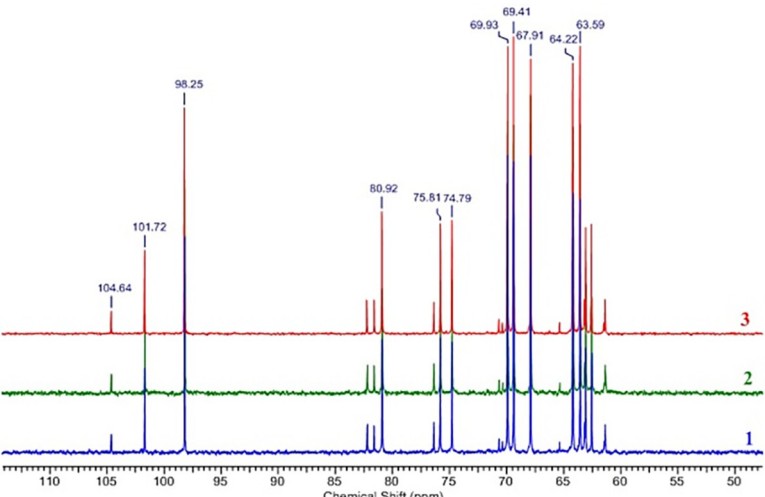

**Fig 6. Superposition of $^{13}$C NMR spectra of HClO$_4$-hydrolyzed samples of EPS *P. polymyxa* 2020 from sucrose medium (1), molasses medium (2) and fructose (3).**

($\geq$99%, Sigma-Aldrich) showed their identity (**Fig 6**) and clearly indicated that the monomer unit of the polysaccharide was fructose.

The $^{13}$C NMR spectrum of the native polysaccharide (**Fig 7**) contained six major signals accounting for a regular glycopolymer, including one anomeric carbon at δ 105.45, two

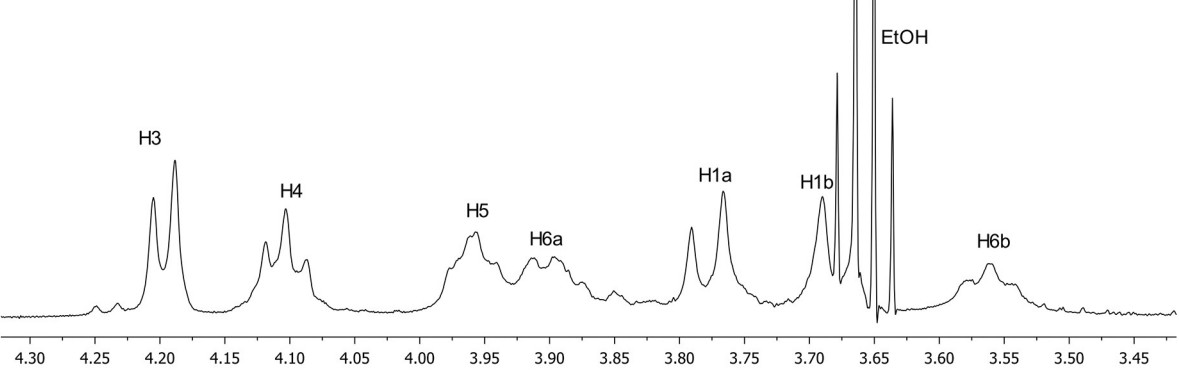

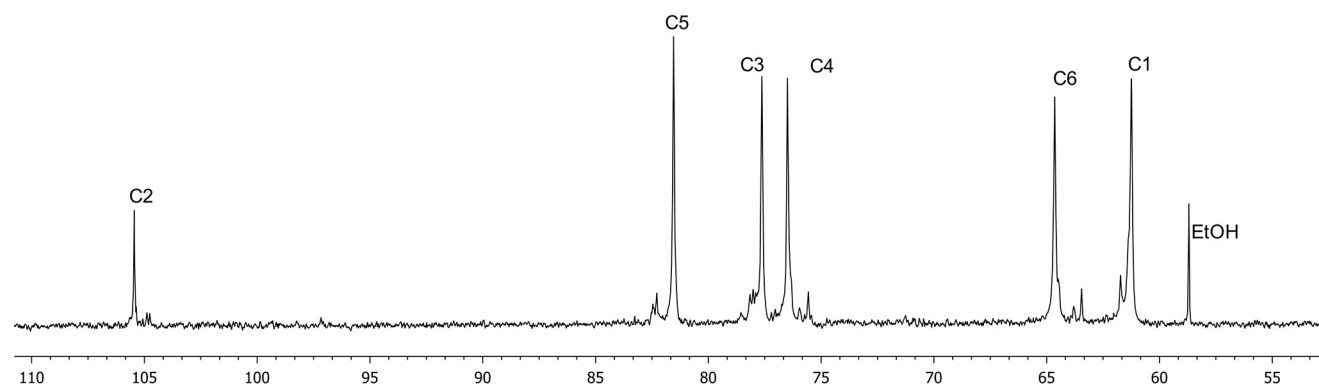

**Fig 7. $^1$H and $^{13}$C NMR spectra of the polysaccharide of *P. polymyxa* 2020 and signal assignment.**

$CH_2OH$ groups, three sugar ring carbon signals typical for a furanose cycle, and minor signals (<10%). The preliminary interpretation of $^{13}C$ NMR chemical shifts by GRASS (Generation, Ranking and Assignment of Saccharide Structures) [66] hosted at the Carbohydrate Structure Database [67] yielded only one match, namely polymeric 2,6-linked β-D-fructofuranose (levan), with linear correlation (LC) between the simulation and the experiment of 1.000, and root-mean-square deviation (RMS) of 0.49 ppm. The next-ranked structural hypothesis had a much worse fit to the experiment (LC 0.994, RMS 1.18 ppm). The top-ranked hypothesis was checked against published NMR spectra of levan [68,69] and exhibited an excellent match, including major and minor signals. The simple and regular signal pattern allowed the complete $^1H$ and $^{13}C$ assignment (**Table 1**) without tracing the atom connectivities via additional NMR experiments.

Thus, it was found by IR and NMR spectroscopy that the exopolysaccharide formed by *P. polymyxa* 2020 is levan, a neutral homopolysaccharide consisting of D-fructopyranose residues linked by β-2→6 bonds.

**Molecular weight.** Levans from the most-studied Gram positive bacterial genera, *Bacillus* and *Paenibacillus*, have molecular weights in the $10^4$ to $10^9$ Da range [70,71], while levan from *B. licheniformis* 8-37-0-1 has a molecular weight of 2.826 x$10^4$ Da [72]. However, the levan produced by *Kozakia baliensis* has an exceptionally high molecular weight of 2.466 x $10^9$ Da [73]. The molecular weights of microbial levans are quite variable depending on the specific producer and the cultivation conditions [74]. For example, *B. subtilis* (natto) produced both low (11 kDa) and high (1800 kDa) molecular weight levans. However, *B. polymyxa* (NRRLB-18475) produced only very high molecular weight levan of approximately $2 \times 10^6$ Da [70,75]. The phenomenon of simultaneous production of high and low molecular weight of levans was observed in *B. subtilis* natto (2.3 x $10^6$ and 7.2 x $10^3$ Da) [19,75] and *B. licheniformis* (6.12 x $10^5$ and 1.1 x $10^4$ Da) [76]. The molecular mass of the EPS produced by *P. polymyxa* 92 in a medium with 10% sucrose was $2.29–1.10 \times 10^5$ Da [33]. The molecular weight of EPS produced by *P. bovis* sp. nov BD3526 on 20% sucrose was $4.8 \times 10^6$ Da [74].

In our work, it was shown that in a culture medium with sucrose at an initial concentration of 50 g $L^{-1}$ and 100 g $L^{-1}$, high molecular weight levan is formed with a molecular weight above 670 kDa (**Fig 8A**). While in a culture medium with sucrose in the initial concentration of 150 g $L^{-1}$ and 200 g $L^{-1}$, a low molecular weight levan of 10–12 kDa is formed. In media with molasses containing sucrose at concentrations of 50 g $L^{-1}$, 100 g $L^{-1}$, 150 g $L^{-1}$, and 200 g $L^{-1}$ high molecular weight levan is formed with a molecular weight above 670 kDa (**Fig 8B**). Indeed, similar effect was previously reported. Relatively high molecular weight levan (range > $10^6$ Da) is produced in sugar beet molasses-based media with 200 g $L^{-1}$ total sucrose, while relatively lower molecular weight levan (in the range of $10^5$ Da) is produced in medium with high concentrations (400 g $L^{-1}$) of sucrose [49].

Thus, the difference in molecular weights between synthesized levans with different sucrose concentrations in the media may indicate a potential controlling mechanism.

**Scanning Electron Microscopy (SEM)/Energy-dispersive X-ray spectroscopy (EDX).** Samples used for structural analysis were prepared by freeze-drying. The SEM showed that the EPS surface has complex petal morphology with rounded macropores (Fig 9). The structure of the sample was similar to that of an aerogel, although it is known that levan cannot be gelled

**Table 1. $^{13}C$ and $^1H$ NMR data of polysaccharide, ppm.**

| Atom | C1/H1a,b | C2 | C3/H3 | C4/H4 | C5/H5 | C6/H6a,b |
|------|----------|-----|-------|-------|-------|----------|
| →6)-β-D-Fru*f*-(2→ $^{13}C$ signal | 61.25 | 105.45 | 77.63 | 76.49 | 81.54 | 64.64 |
| →6)-β-D-Fru*f*-(2→ $^1H$ signal | 3.77 (d), 3.68 (d) | - | 4.20 (d) | 4.10 (t) | 3.96 (m) | 3.90 (m), 3.56 (t) |

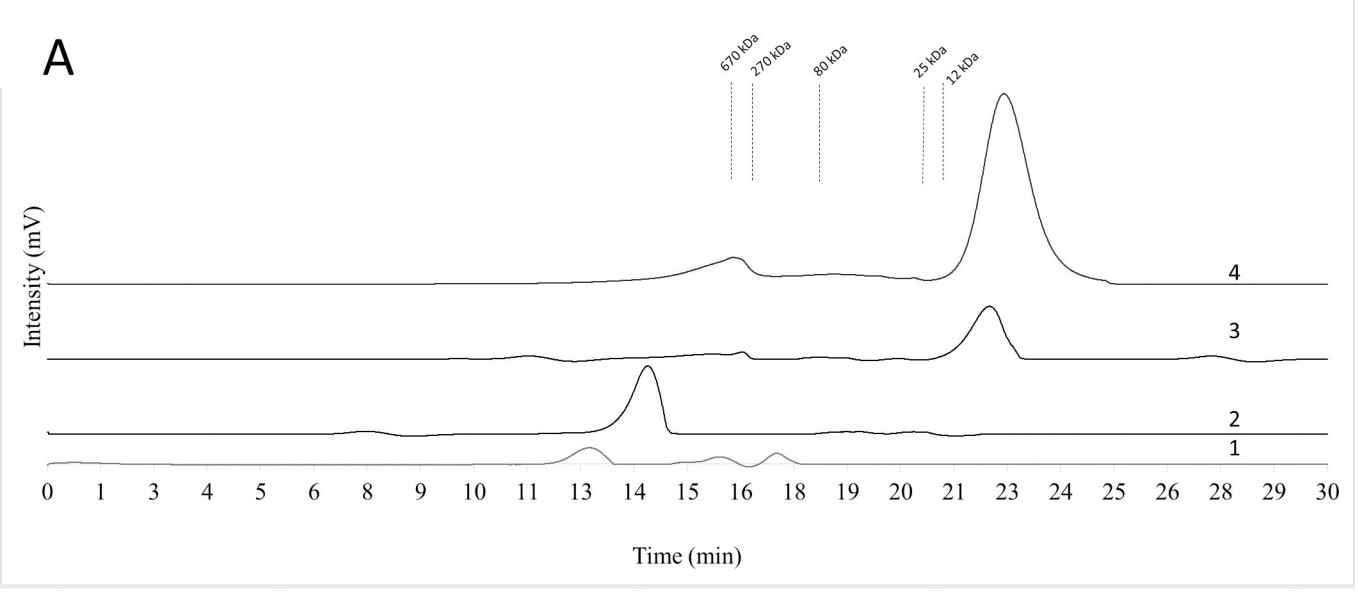

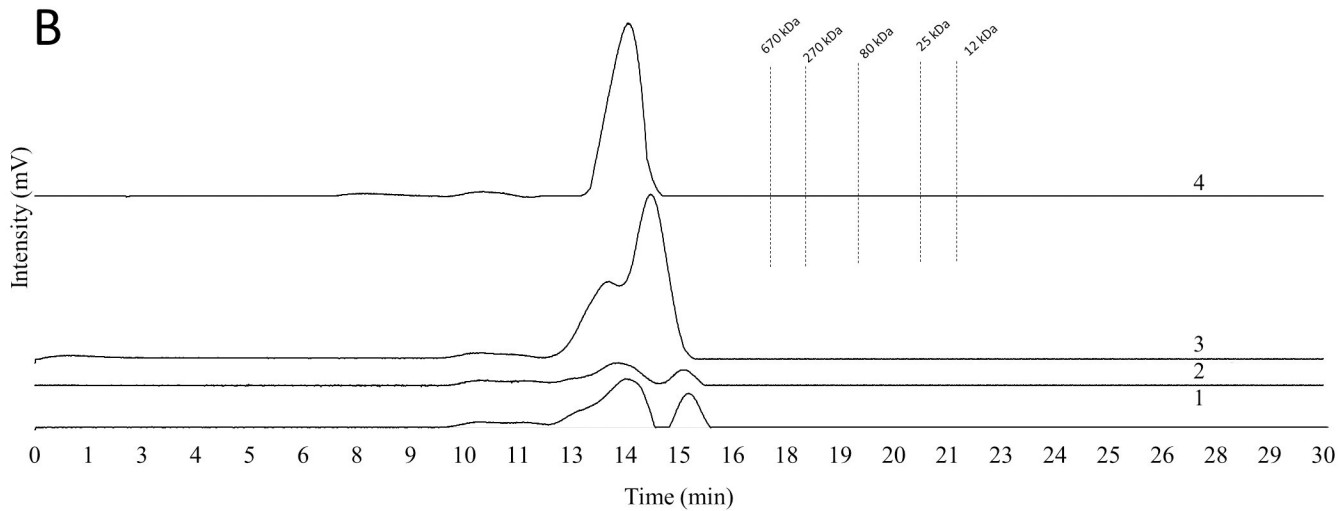

**Fig 8.** Molecular weight distribution of the EPSs produced by *P. polymyxa* 2020 in a sucrose (A) and molasses (B) media with sucrose at an initial concentration of 50 g L$^{-1}$ (1), 100 g L$^{-1}$ (2), 150 g L$^{-1}$ (3) and 200 g L$^{-1}$ (4) The dashed lines indicate the retention times of the dextran standard compounds used.

without a surface modification. Aerogels are known as highly porous materials with a huge internal surface area [77]. Two methods usually applied to form aerogels are (1) supercritical drying or (2) freeze-drying. In both cases, the structure of the material is preserved during drying. An analysis of the sample composition was performed using energy-dispersive X-ray spectroscopy (EDX) and the elements present in the sample were analyzed by (**S5 Fig**). The results demonstrated strong C and O signals and allowed the identification of 57.49% carbon and 41.77% oxygen corresponding to groups present in the levan structure.

**Thermogravimetric analysis (TGA).** The stability and longevity of polymers are usually measured by thermogravimetric analysis. One important characteristic of EPS is its thermostability, especially for high-temperature applications. Indeed, levan can be fully thermally processed through molding and extrusion methods [4]. Therefore, we conducted a

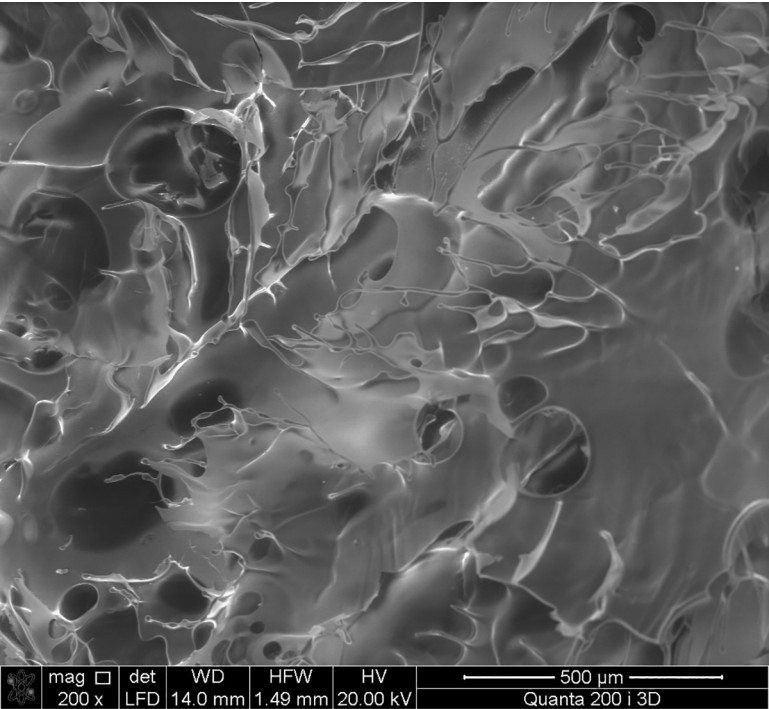

**Fig 9. SEM image of EPS of *P. polymyxa* 2020.**

thermogravimetric study of the EPSs produced by *P. polymyxa* 2020 during growth in sucrose and molasses media (**Fig 10**). The thermal weight loss of the EPS started at 57°C, and when the treatment reached 200°C, the EPS lost about 15% of its mass. Thermogravimetric analysis demonstrated that levan begin to degrade at 200°C, and by the time the temperature reached 300°C about 60% of the EPS mass has been lost.

## Cytotoxicity

To investigate the effect of levan produced by *P. polymyxa* 2020 from a sucrose and molasses medium, cytotoxicity studies were performed to test the proliferation of a mouse fibroblast cell

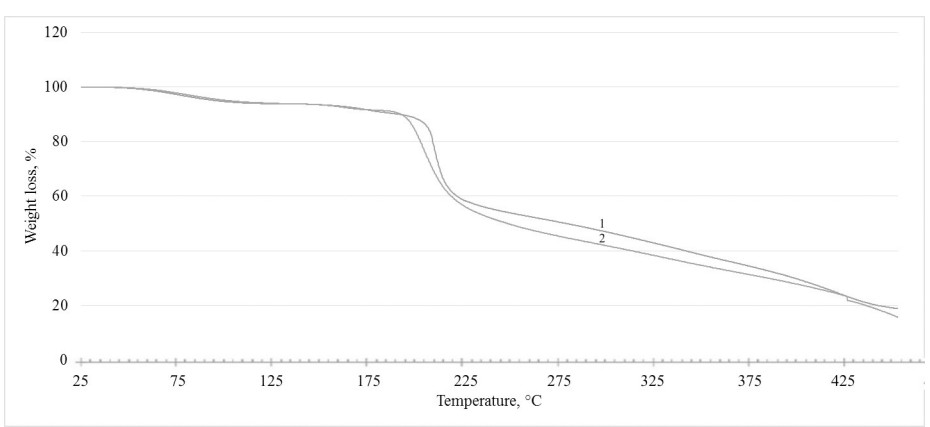

**Fig 10. Thermogravimetric analysis of EPS of *P. polymyxa* 2020 from sucrose medium (1) and EPS of *P. polymyxa* 2020 from molasses medium (2).**

culture L929. The viability of L929 cells was evaluated by the MTT assay (**S4 Fig**). All the materials tested showed very low toxicity. No reduced cell viability was seen during incubation with levan produced by *P. polymyxa* 2020 either in a sucrose or a molasses media. Thus, EPS does not inhibit the proliferation of L929 cells. We also observed that the morphology of L929 cells was not affected by levan produced by *P. polymyxa* 2020 in a sucrose and molasses media. The cells were monitored using an inverted optical microscope and the morphologies of untreated L929 cells were similar to those that had been treated with EPS. These results indicate that levan produced by *P. polymyxa* 2020 is a promising source of functional biomaterials for a wide range of cellular applications, including biomedicine.

## Molecular and biochemical analysis of *P. polymyxa* 2020 levan-synthesizing operon and SacB, SacC gene products

The product of the *sacB* gene is known as levansucrase and was originally cloned from *Bacillus subtilis* into *E.coli* [78]. The *sacB* gene is toxic for *E.coli* when grown on sucrose, therefore it found application as a negative selection marker for allele exchange-eviction-knockout mutagenesis that enables the construction of directed, unmarked mutations in gram-negative bacteria [79].

Previously we demonstrated, that *P. polymyxa 2020* indeed is a producer of levan. Therefore, we performed a *P. polymyxa 2020* genome search for a *sacB* gene known to encode levansucrase using NCBI tblastn algorithm with the SacB protein sequence from *Bacillus amyloliquefaciens* as a query. Indeed, we found a single genome hit at position 1,072,464–1,073,849 bp (**Fig 11A**). Bioinformatic analysis, using Geneious Prime Software, revealed the presence of a putative levan producing operon in the genomes of both *P. polymyxa* strains that potentially encode 8 ORFs (**Fig 11B**). Blast analysis indicated that ORF530 is the putative *sacC* gene encoding a glycoside hydrolase family 32 protein, ORF502 is a putative *sacB* encoding a glycoside hydrolase family 68 protein and several potential transporter proteins (ORF504-MFS transporter, ORF519-ABC transporter ATP-binding protein, ORF203-ECF transporter S component, ORF262-energy-coupling factor transporter transmembrane protein EcfT, ORF196-GNAT family N-acetyltransferase and ORF101-hypothetical protein). The original attempt to clone the entire Sac operon in *E. coli* failed. We were able to amplify a 9,690 bp fragment using P5-P6 primers, but following colony screening only deletion variants were found.

To quickly test the activity of the putative products of the *sacB* and *sacC* genes appropriate ORFs were PCR amplified and cloned into a T7 vector, pSAPv6. The resulting plasmids pPpo-*SacB* and pPpo*SacC* were used for *in vitro* expression in the PURExpress system and these products were detected on a 10–20% SDS PAGE gel (**Fig 11C**). The sequence estimated molecular weight of SacB protein is 55,709.83 Da and SacC protein is 59,788.94 Da. However, the SDS PAGE analysis indicated that the SacB protein runs slower than the SacC protein, indicating either an unusual structure or some additional component(s) bound to the SacB protein. Also negative charge of SacB protein may contribute to slow migration on a gel. Estimated isoelectric point of SacB protein is 5.194 (-10.741 charge at pH7.0) and isoelectric point of SacC protein is 8.869 (7.714 charge at pH7.0) (https://web.expasy.org/compute_pi/). Both proteins run between 55 and 72 kDa protein markers (**Figs 11C** and S7).

Levansucrase enzyme activity was assessed by qualitative determination of glucose, fructose monosaccharides and oligosaccharides (kestose) derived from sucrose after incubation with PURExpress SacB and SacC proteins. The samples were analyzed by hydrophilic interaction liquid chromatography coupled with electrospray ionization mass spectrometry (UPLC-HILIC-MS). The deprotonated negative ions [M-H]$^-$ of glucose and/or fructose (*m/z* 179) and sucrose (*m/z* 341) were found to be dominant in the respective ESI mass spectra. These parent ions were used to detect the analyzed sugars from full scan and selective ion recording (SIR)

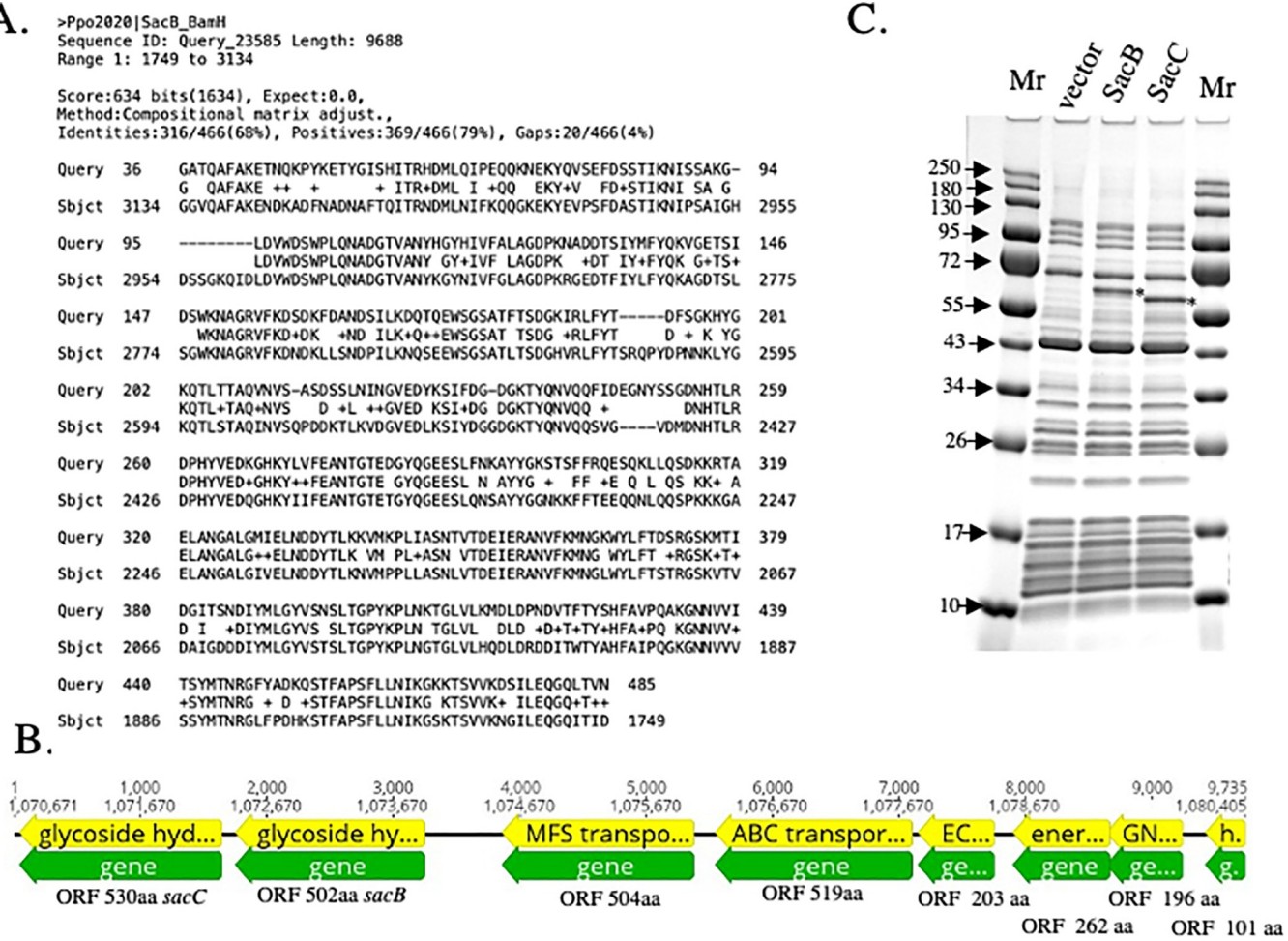

**Fig 11.** Blast hit on *P. polymyxa* 2020 chromosome with SacB protein from *Bacillus subtilis* as a query (A). Putative Sac operon from *P. polymyxa* 2020 (B) and 10–20%SDS PAGE of PURExpress extracts of pSAPv6 (vector), pPpo2020_*sac*B (SacB) and pPpo2020_*sac*C (SacC). Position of recombinant SacB and SacC proteins indicated by * (C).

chromatograms. Incubation of sucrose (retention time– 9.5 min) with putative SacB protein resulted in efficient, albeit incomplete, hydrolysis of the substrate, whereas very little sucrose hydrolysis was observed after incubation with SacC (**Fig 12A**).

The fructose and glucose peaks (retention times– 4.5 and 5.5 minutes, respectively) were detected in the SacB reaction at *m/z* 179 (**Fig 12B**). Other peaks are also apparent, suggesting the presence of other saccharides. Traces of fructose and minor glucose peak were also detected after incubation with SacC (**Fig 12B**). These results indicate that both enzymes catalyze hydrolysis of sucrose, but with different efficiencies. At least one short-chain oligosaccharide (trisaccharide, most probably kestose) was also observed in the SacB reaction (**Fig 12C**). Therefore, both hydrolytic and transferase activities of SacB were detected, suggesting that SacB is a levansucrase type enzyme.

## Conclusion

In this study, we have obtained a highly productive strain, *Paenibacillus polymyxa* 2020, first isolated from wasp honeycombs, capable of producing a greater amount of levan compared to

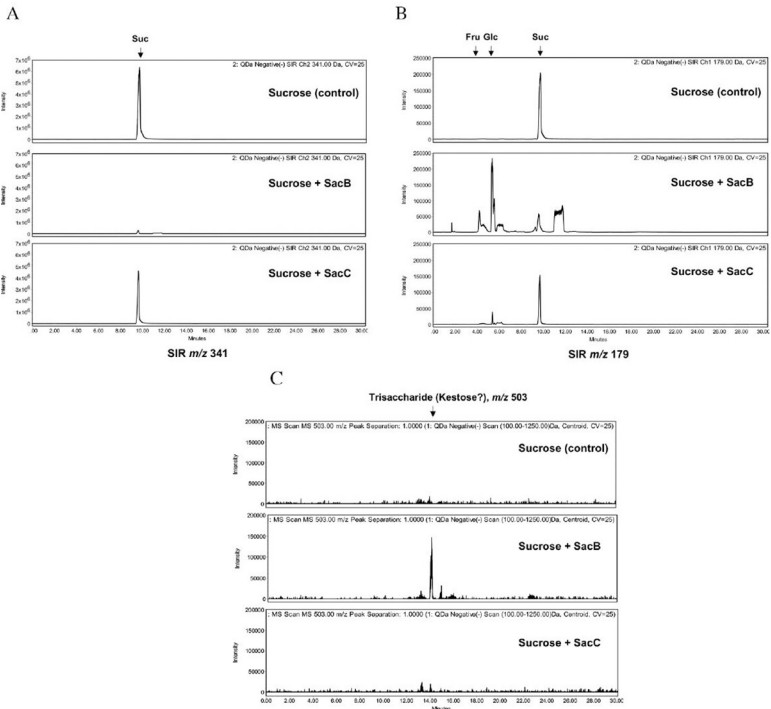

**Fig 12.** SIR profiles of the sucrose (*m/z* 341) (A) and glucose/fructose (*m/z* 179) (B), observed after sucrose incubation with SacB, SacC proteins. Due to in-source fragmentation, the sucrose fragmentation product ion is also observed at *m/z* 179 in panel B. Extracted ion chromatograms of trisaccharide (*m/z* 503) (C) from full scan MS in negative mode.

the known *P. polymyxa* strains. Whole genome sequencing and methylome analysis of *P. polymyxa* 2020 were performed. Bioinformatic analysis identified a putative levan synthetic operon. S*acC* and *sacB* genes have been cloned and their products identified as glycoside hydrolases and levansucrase. The highest levan yield of 68.0 g L$^{-1}$ was obtained on a molasses medium with a total sugar concentration of 200 g L$^{-1}$ at 72 h. To the best of our knowledge, this is the highest yield of levan on molasses-based medium reported so far. The highest yield of EPS in sucrose medium was 53.78 g L$^{-1}$ with 150 g L$^{-1}$ sucrose at 96 h. This is the maximum levan yield on sucrose-containing media in comparison with the literature data known for *P. polymyxa*. Besides, in comparison to the media with sucrose, in the molasses medium, the maximum production of polysaccharide was achieved over a shorter time interval (48–72 h). Therefore, molasses has been shown to be a cheaper and renewable alternative to sucrose for the cost-effective levan production by *P. polymyxa* 2020. Purified EPS was identified as a linear β-(2→6)-linked fructan (levan) by the Fourier transform infrared (FT-IR) and nuclear magnetic resonance (NMR) spectra analysis. Structural analysis of both levan polymer produced from sucrose and molasses by FT-IR, NMR, SEM, HPSEC and TGA indicated high level of similarity. The thermal properties of levan indicated it to be easily and fully thermally processed through traditional moulding and extrusion techniques. Levan polymer produced by *P. polymyxa* 2020 from sucrose and molasses showed low toxicity and high biocompatibility. Therefore, the present work highlighted the potential of the *P. polymyxa* 2020 for the cost-effective industrial production of a levan-type EPS and the obtaining of functional biomaterials based on it for a wide range of applications, including biomedicine.

## Supporting information

**S1 Fig.** Biomass accumulation (A) and the changes in pH (B) during growth of *P. polymyxa* 2020 in a sucrose medium containing sucrose at concentrations 50 g L$^{-1}$ (1), 100 g L$^{-1}$ (2), 150 g L$^{-1}$ (3), 200 g L$^{-1}$ (4).
(TIF)

**S2 Fig.** Biomass accumulation (A) and the changes in pH (B) during growth of *P. polymyxa* 2020 in molasses medium containing sucrose at concentrations 50 g L$^{-1}$ (1), 100 g L$^{-1}$ (2), 150 g L$^{-1}$ (3), 200 g L$^{-1}$ (4).
(TIF)

**S3 Fig.** Sucrose (1), glucose (2) and fructose (3) content during growth of *P. polymyxa* 2020 in a culture medium with sucrose at an initial concentration of 50 g L$^{-1}$ (A), 100 g L$^{-1}$ (B), 150 g L$^{-1}$ (C) and 200 g L$^{-1}$ (D) for 4 days (by HPLC).
(TIF)

**S4 Fig.** Sucrose (1), glucose (2) and fructose (3) content during growth of *P. polymyxa* 2020 in a molasses medium with sucrose at an initial concentration of 50 g L$^{-1}$ (A), 100 g L$^{-1}$ (B), 150 g L$^{-1}$ (C) and 200 g L$^{-1}$ (D) for 4 days (by HPLC).
(TIF)

**S5 Fig. Results of EDX analysis of EPS.**
(DOC)

**S6 Fig.** Cell viabilities (A) and cell morphologies (B) of the mouse fibroblast cell culture L929 treated with levan produced *P. polymyxa* 2020 in a sucrose (Ls) and molasses (Lm) media.
(TIF)

**S7 Fig. 4–20% SDS PAGE unadjusted and uncropped image of gel.** Lanes 1, 5, 7, 11–10 μl of color prestained protein standards broad range (10-250kDa) (NEB #P7719); lanes 2, 8–2.5 μl of PurExpress extract from empty vector pSAPv6; lanes 3, 9–2.5 μl of PurExpress extracts from pPpo2020_*sac*B template plasmid; lanes 4, 10–2.5 μl of PurExpress extracts from pPpo2020_-*sac*C template plasmid; lanes 6, 12–2.5 μl of PurExpress extracts from pPpo2020_sacBC template plasmid; plasmids purified from ER2683 *E.coli* strain (lanes: 2, 3, 4, 6); plasmids purified from ER3081 *E.coli* strain (lanes 8, 9, 10, 12).
(TIF)

**S1 Table. Bacterial strains, plasmids and oligonucleotides.**
(XLSX)

## Acknowledgments

We thank Peter Weigele with his help on bioinformatic analysis on Geneious Prime Software.

## Author Contributions

**Conceptualization:** Elena V. Liyaskina, Alexey Fomenkov, Saulius Vainauskas, Victor V. Revin.

**Data curation:** Elena V. Liyaskina, Nadezhda A. Rakova, Alevtina A. Kitykina, Valentina V. Rusyaeva, Philip V. Toukach, Alexey Fomenkov, Saulius Vainauskas, Victor V. Revin.

**Formal analysis:** Nadezhda A. Rakova, Alevtina A. Kitykina, Valentina V. Rusyaeva, Philip V. Toukach, Alexey Fomenkov, Saulius Vainauskas, Victor V. Revin.

**Funding acquisition:** Victor V. Revin.

**Investigation:** Alevtina A. Kitykina.

**Methodology:** Nadezhda A. Rakova, Alevtina A. Kitykina, Philip V. Toukach, Saulius Vainauskas.

**Project administration:** Victor V. Revin.

**Resources:** Elena V. Liyaskina, Saulius Vainauskas.

**Validation:** Nadezhda A. Rakova, Alevtina A. Kitykina, Philip V. Toukach.

**Visualization:** Nadezhda A. Rakova, Alevtina A. Kitykina.

**Writing – original draft:** Elena V. Liyaskina.

**Writing – review & editing:** Alexey Fomenkov, Richard J. Roberts.

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
