## [Decision Letter · Decision Letter 0]

18 Feb 2021

PONE-D-20-40095

Production and сharacterization of the exopolysaccharide from novel strain Paenibacillus polymyxa 2020

PLOS ONE

Dear Dr. Formenkov,

Thank you for submitting your manuscript to PLOS ONE. After careful consideration, we feel that it has merit but does not fully meet PLOS ONE’s publication criteria as it currently stands. Therefore, we invite you to submit a revised version of the manuscript that addresses the points raised during the review process.

We look forward to receiving your revised manuscript.

Kind regards,

Dharam Singh

Academic Editor

PLOS ONE

Journal Requirements:

2.We note that you have indicated that data from this study are available upon request. PLOS only allows data to be available upon request if there are legal or ethical restrictions on sharing data publicly. For more information on unacceptable data access restrictions, please see http://journals.plos.org/plosone/s/data-availability#loc-unacceptable-data-access-restrictions.

3.We note that the grant information you provided in the ‘Funding Information’ and ‘Financial Disclosure’ sections do not match.

4. Please upload a copy of Figure 14, to which you refer in your text on page 22. If the figure is no longer to be included as part of the submission please remove all reference to it within the text.

5.Thank you for stating the following in the Competing Interests section:

"A.F, S.V. and R.J.R. work for New England Biolabs, company that sells research reagents, including restriction, glyco enzymes and DNA methyltransferases, to the scientific community. "

We note that one or more of the authors are employed by a commercial company: New England Biolabs Inc.,

6.Thank you for submitting the above manuscript to PLOS ONE. During our internal evaluation of the manuscript, we found significant text overlap between your submission and the following previously published works, some of which you are an author.

Introduction section:

- https://www.sciencedirect.com/science/article/abs/pii/S0141813018332847?via%3Dihub

Results, Discussion, and Conclusion sections:

- https://microbialcellfactories.biomedcentral.com/articles/10.1186/s12934-016-0603-7

- https://mra.asm.org/content/8/32/e00657-19

- https://pubmed.ncbi.nlm.nih.gov/31879785/

- https://www.sciencedirect.com/science/article/abs/pii/S0141813018332847?via%3Dihub

- https://link.springer.com/article/10.1007/BF01569633

- https://www.sciencedirect.com/science/article/abs/pii/S0141813016308777?via%3Dihub

- https://www.sciencedirect.com/science/article/abs/pii/S014486171630114X?via%3Dihub

- https://www.frontiersin.org/articles/10.3389/fbioe.2020.603407/full

Please revise the manuscript to rephrase the duplicated text, cite your sources, and provide details as to how the current manuscript advances on previous work. Please note that further consideration is dependent on the submission of a manuscript that addresses these concerns about the overlap in text with published work.

Reviewers' comments:

Reviewer's Responses to Questions

**Comments to the Author**

1. Is the manuscript technically sound, and do the data support the conclusions?

Reviewer #1: Yes

Reviewer #2: Partly

2. Has the statistical analysis been performed appropriately and rigorously? 

Reviewer #1: N/A

Reviewer #2: N/A

3. Have the authors made all data underlying the findings in their manuscript fully available?

Reviewer #1: Yes

Reviewer #2: Yes

4. Is the manuscript presented in an intelligible fashion and written in standard English?

Reviewer #1: Yes

Reviewer #2: Yes

5. Review Comments to the Author

Reviewer #1: This study reports about isolation of Paenibacillus polymyxa 2020 from wasp honeycombs capable of producing levan. Whole genome sequencing and methylome analysis of P. polymyxa 2020 were performed and bioinformatic analysis identified putative levan synthetic operon. sacC and sacB genes were further cloned and their products identified. To reduce the production cost inexpensive substrate molasses was investigated for EPS production. Manuscript needs following changes to improve it further.

Comments:

1. Remove word novel from title as many Paenibacillus strains already isolated and reported for different applications.

2. In introduction part add information related to different exopolysaccharides, microorganisms, and their applications. Also add information related to market potential of levan and other EPS.

3. Line 138: Why author decided to use 10% v/v inoculum? Why author didn’t try to optimize various culture conditions for optimum growth and EPS production, as this was a new isolate, and these parameters may help to increase production.

4. What are other components present in EPS? As EPS is composed of carbohydrates, proteins, organic acids etc.

5. Line 339: P. polymyxa 2020 showed a high content of EPS in comparison to other strain at almost same concentration of sucrose. Compare in terms of culture conditions also (pH and T) used in this study and other study to provide better information to readers.

6. Provide figure number to all the figures.

7. Remove grid lines from the figures.

8. Line 99: C/N and pH plays important role in EPS production, while in this study no such experiment included. Need to perform experiment and include results.

Reviewer #2: The paper is interesting in describing a new strain with relevant biosynthetic pathways to exopoysaccharide production, and the ability to achieve high titer of EPS. The authors run a wide array of characterization experiments, from a biomolecular and biological side to a more chemical-structure analyses.

However the manuscript fails to correctly describe in detail the experimental part.

I think there is a need to improve the methodologies section few examples are following:

Improvement of fermentation descriptions, give detains on growth.

explain how they can purify and isolate EPS with such a simple 2 step process (comparison to other EPS, contamination of polysaccharides from teh media...etc.)

Clarify why they use high temperature drying

improve SEM analyses description

compared their limitation in NMR to eventually previously reported spectroscopic analyses of heteropolysaccarides like hyaluronic acid, or other GAGs.;

Explain the meaning of the cytotoxicity study with respect to a proposed application;

e) describe the specific interest in the molecular determinant of the biosynthetic pathways, since they report already a very high production (and productivity) when compared to other microbial polysaccharides.

Minor points are also highlighted below:The shake flasks growth, although a simple and routine approach should be better described here (biomass density achieved, growth curve, or any othe information like change in pH during growth.

If the process is a fully aerobic one it seems that the inoculums of 100 mL medium in a 250 mL flask is not ensuring a sound gas distribution, so is limiting oxygen transfer, this especially in relation to the increase n velocity during growth.

In Figure 2 the production of levan is presented without any information on the biomass growth. Since the authors are using a very high sugar content (either a disaccharide or a complex mixtures ..also containing oligo saccharides) the osmotic pressure will be different in the various condition and thus I will expect a difference in growth rate and then of Levan production.

The reference at time zero (for zero amount of levan ) was quantified on the medium of this was an assumption of the authors? Please specify.

The downstream procedure described is very simple, I argue it is very difficult to separate pure EPS only using precipitation and dialysis , did the author apply th e same procedure to the fermentation broth to analyse similar residues directly coming from the complex components (e.g. yeast extract or other) .

Fig 10. SEM images of EPS of P. polymyxa 2020 at a various magnification- please improve description in the legend

SEM micrograph should be better explained in the text (how the films were obtained, and with procedeure before metalization. It is not clear why a porosity should be found if not for very low molecular weight components (like salts) that may leave the film if washed . The authors should clarify this point0

Figure 12 is very confusing, and low quality

I appreciated the effort of the research group to address all together key aspects of production and purification of microbial polysaccharides, but such an extensive overview shoudl not limit the relevant information to sustain the quality of the research and the reproducibility of the results.

6. PLOS authors have the option to publish the peer review history of their article (what does this mean?). If published, this will include your full peer review and any attached files.

Reviewer #1: **Yes: **Shashi Kant Bhatia

Reviewer #2: No

---

## [Author Response · Author response to Decision Letter 0]

26 Mar 2021

PONE-D-20-40095

Production and сharacterization of the exopolysaccharide from strain Paenibacillus polymyxa 2020

PLOS ONE

Authors’ response: Files were reformatted according to PLOS One style.

2.We note that you have indicated that data from this study are available upon request. PLOS only allows data to be available upon request if there are legal or ethical restrictions on sharing data publicly. For more information on unacceptable data access restrictions, please see http://journals.plos.org/plosone/s/data-availability#loc-unacceptable-data-access-restrictions.

Authors’ response: All appropriate data were deposited into the NCBI database. Lines 315-321 have all appropriate references. There are no restrictions on the data. 

Authors’ response: 

This work was supported by the Ministry of Science and Higher Education of the Russian Federation, grant number FZRS-2020-0003. New England Biolabs provided support in the form of salaries for authors A.F, S.V. and R.J.R. The specific roles of these authors are articulated in the ‘author contributions’ section. The funders had no role in study design, data collection and analysis, decision to publish, or preparation of the manuscript.'

Please remove second reference to grant 18-14-00098 from a system, because I cannot update it

"The instrumental and computational NMR investigation was carried out within the Glycoinformatics project funded by Russian Science Foundation, grant 18-14-00098".

4. Please upload a copy of Figure 14, to which you refer in your text on page 22. If the figure is no longer to be included as part of the submission please remove all reference to it within the text.

Authors' response: All figures were uploaded. Their numbering was changed. New figures were added.

5.Thank you for stating the following in the Competing Interests section:

"A.F, S.V. and R.J.R. work for New England Biolabs, company that sells research reagents, including restriction, glyco enzymes and DNA methyltransferases, to the scientific community. "

We note that one or more of the authors are employed by a commercial company: New England Biolabs Inc.,

Authors' response: 

The authors have read the journal's policy and have the following competing interests: A.F, S.V. and R.J.R. are paid employees of New England Biolabs (https://www.neb.com/). There are no patents, products in development or marketed products associated with this research to declare. This does not alter our adherence to PLOS ONE policies on sharing data and materials.'

6.Thank you for submitting the above manuscript to PLOS ONE. During our internal evaluation of the manuscript, we found significant text overlap between your submission and the following previously published works, some of which you are an author.

Introduction section:

- https://www.sciencedirect.com/science/article/abs/pii/S0141813018332847?via%3Dihub

Results, Discussion, and Conclusion sections:

- https://microbialcellfactories.biomedcentral.com/articles/10.1186/s12934-016-0603-7

- https://mra.asm.org/content/8/32/e00657-19

- https://pubmed.ncbi.nlm.nih.gov/31879785/

- https://www.sciencedirect.com/science/article/abs/pii/S0141813018332847?via%3Dihub

- https://link.springer.com/article/10.1007/BF01569633

- https://www.sciencedirect.com/science/article/abs/pii/S0141813016308777?via%3Dihub

- https://www.sciencedirect.com/science/article/abs/pii/S014486171630114X?via%3Dihub

- https://www.frontiersin.org/articles/10.3389/fbioe.2020.603407/full

Please revise the manuscript to rephrase the duplicated text, cite your sources, and provide details as to how the current manuscript advances on previous work. Please note that further consideration is dependent on the submission of a manuscript that addresses these concerns about the overlap in text with published work.

Authors' response:

 I would like to apologize for the presence of duplicated paragraphs in our original submission. The original manuscript was written in Russian with small pieces of paragraphs written in English. While making the full translation I did not realize that these pieces were merely exact language taken from cited articles. We rephrased these duplicated paragraphs as shown below. 

The duplicated text from Grady et al., 2016. Revised to Line 281-283

Members of the Paenibacillus genus have been isolated from different environments. While many of the species are relevant to humans, animals, and plants, the majority are found in soil, often associated with plant roots to promote growth They can be to improve agriculture [31].

The duplicated text from Revin et al., 2020. Revised to Line 228-242

The L929 mouse fibroblast cell culture used in this study was obtained from the tissue culture collection of the D.I. Ivanovsky Institute of Virology, Russia. The cells were cultured in Dulbecco’s modified Eagle's medium (DMEM) (Paneko, Russia) containing 10% fetal bovine serum (FBS) (HyClone, USA) in the presence of penicillin and streptomycin under standard conditions: 5% CO2 atmosphere, t = 37°C, 5% humidity in an MCO-170M incubator (Sanyo, Japan). Cells in exponential growth phase were dispened into a 96-well plate (5 × 103 cells/well). After 24 hr, fresh medium, containing the EPSs was added to the 96 wells and incubated for a further 24hr. Wells without EPSs were used as a control. The morphological structure of the cells were monitored using an inverted optical microscope (Micromed, Russia). The MTT assay was used to measure cytotoxicity of the EPSs. The medium was replaced with a fresh one containing 5 mg/mL dimethyl thiazolyl diphenyl (MTT). After 4 h incubation time the medium was removed and 150 μL DMSO was added. The optical density was measured on a microplate reader EFOS 9305 (Russia) at a wavelength of 492 nm with a reference wavelength of 620 nm. Cell viability was recorded as the ratio of the optical density of the sample to the control and expressed as a percentage.

The duplicated text from H.Mankai et.al., 2019. Revised to Line 151-164

 SMRTbell libraries were prepared using a modified PacBio protocol adapted for NEB reagents. Genomic DNA samples were sheared to an average size of ~ 6-10 kb using the G-tube protocol (Covaris; Woburn, MA, USA), treated with FFPE, end repaired, and ligated with hairpin adapters. Incompletely formed SMRTbell templates were removed by digestion with a combination of exonuclease III and exonuclease VII (New England Biolabs; Ipswich, MA, USA). The qualification and quantification of the SMRTbell libraries were made on a Qubit fluorimeter (Invitrogen, Eugene, OR) and a 2100 Bioanalyzer (Agilent Technologies, Santa Clara, CA). SMRT sequencing was performed using a PacBio RSII (Pacific Biosciences; Menlo Park, CA, USA) based on standard protocols for 6-10 kb SMRTbell library inserts. Sequencing reads were collected and processed using the SMRT Analysis pipeline from Pacific Biosciences

(http://www.pacbiodevnet.com/SMRT-Analysis/Software/SMRT-Pipe) [53].

Next-generation SMRT sequencing technology from Pacific Biosciences Inc. allowed the assembly of a complete circular genome. It also enabled the determination of the m6A and m4C modified motifs. [54-56].

The duplicated text from Grady et al., 2016. Revised to Line 323-329

The genome sequences of 212 strains of Paenibacillus, that represent 82 different species are available [31]. Genome sizes range from 3.02 Mb for P. darwinianus Br, isolated from Antarctic soil [60] to 8.82 Mb for P. mucilaginosus K02, implicated in silicate mineral weathering [61] and the number of genes varies from 3,064 for P. darwinianus Br to 8,478 P. sophorae S27. The insect pathogens P. darwinianus, P. larvae and P. popilliae have smaller genomes from 4.51 and 3.83 Mb, respectively, perhaps reflecting their niche specialization. The GC content of Paenibacillus DNA ranges from 39 to 59 % [62].

The duplicated text from J.R Horton et al., 2020. Revised to Line 285-322

A 6 kb SMRTbell library was sequenced using C4-P6 chemistry and run on 4 SMRT cells with a 360-minute collection time. 1.6 Gb of data were collected from 55,864 sequencing reads with 1,705 bp mean subread lengths. This sequence was assembled de novo using the HGAP_Assembly.3 version 2.3.0 with default quality and read length parameters and polished once using Quiver. The original polished assembly generated 125 linear contigs (the sum of the contigs equaled 5,907,830 bp) with N50 equal to 104,550bp. BLAST analysis at NCBI of the 125 contig assembly of Paenibacillus polymyxa 2020 identified the strongest hits as having 99.5% homology to AFOX01 a shotgun Miseq-based sequence assembly of P. polymyxa DSM36/ATCC842 [42]. 

P. polymyxa DSM36 native genomic DNA (2�g) was obtained directly from the DSMZ collection and a 15kb SMRT bell library was prepared according to the Pacific Biosciences 10-20 kb protocol including additional separation using a Blue Pippin and sequenced with C4-P6 chemistry on 2 SMRT cells. One library consisted of non-size selected (15 kb) DNA and one contained size-selected (17kb) DNA and a 360-minute collection time was used for each sample on the RSII PacBio sequencer. 28,951 sequencing reads with 12,000 bp mean subread lengths, yielded 0.35 Gb of data. This was de novo assembled using the HGAP_Assembly.3 version 2.3.0 with default quality and read length parameters and polished 3 times using Quiver. The polished assembly generated 2 closed circular genomic elements of 5,919,686 bp with 45.08 % GC content for the main chromosome and 45,518 bp with 41.76% GC for the plasmid pPpo45. The assembled sequences were submitted to the NCBI Prokaryotic Genomes Annotation Pipeline (PGAP) [57, 58]. 

A re-run of the original reads from P. polymyxa 2020 against the genomic reference sequence of P. polymyxa DSM36 and polishing 5 times using Quiver demonstrated that 97.7% of the P. polymyxa 2020 reads mapped to the P. polymyxa DSM36 reference and generated a final genome with 99.99 % homology between the two strains.

The same m6A modified DNA motif, CNAGNNNNNTTGK, was detected in both P. polymyxa 2020 and DSM36 strains and can be assigned to a Type I restriction-modification system Ppo36I in the DSM36 strain and Ppo2020I in the 2020 strain (Fig 1), consistent with the notion that these two P. polymyxa strains are essentially identical. All data have been deposited at NCBI and in REBASE [59].

Nucleotide sequence accession numbers

The complete genome sequence of P. polymyxa DSM36 was deposited in NCBI and given the accession numbers: CP049783-CP049784. The original sequence reads also have been deposited at NCBI under SRA SRR11271650 and SRR11271651.Biosample : SAMN14247784. The complete genome sequence of P. polymyxa 2020 also is available in GenBank with the accession numbers: CP049598-CP049599. The original sequence reads have been deposited at NCBI under SRA: SRR11236808; SRR11236809; SRR11236810; SRR11236811. Biosample: SAMN14247689. Data for both strains are available under Bioproject: PRJNA609344

The duplicated text from Gojgic-Cvijovic et al., 2018. Revised to Line 348-352

The use of sugar-based raw materials as nutritive substances can greatly help to reduce the production cost of microbial polysaccharides [49]. Molasses, a food industry by-product of cane or beet sugar production, contains up to 80% of dry matter, of which sucrose represents ~48%. Molasses also contains amino acids, organic acids and their salts, betaine, mineral compounds, and vitamins

The duplicated text from Gojgic-Cvijovic et al., 2018. Revised to Line 384-391

 The pH changes during fermentation on sucrose and on molasses media were compared (S1 Fig(B), S2 Fig) (B). In sucrose medium, a rapid drop of pH occurred during the first 24 h due to acid formation, and afterwards, the pH remained stable in the range of 5.0 - 5.2 (with an initial sucrose concentration of 50 g L-1) or 4.3 - 4.8 (with initial sucrose concentrations of 100, 150 and 200 g L-1) until the end of fermentation (S1 Fig (B)). 

The pH, in molasses medium, was only slightly reduced and ranged from 5.79 to 6.0 throughout the entire fermentation (S2 Fig (B)). Therefore, pH curves were in accordance with previously reported data [49, 75].

The duplicated text from Han, Watson, 1992. Revised to Line 394-399

 The bacterial conversion of sucrose into levan leads to glucose accumulation, as shown by the decrease in sucrose levels and the accompanying increase in levan and glucose levels in the growth medium. A small amount of fructose was also detected. The sucrose level dropped and levan started to appear during 24 h fermentation; thereafter, the sucrose level rapidly decreased as levan increased. Levan levels peaked at 4 days and reached about 36% when grown on 15% sucrose medium, constituting a 47% theoretical yield based on the available fructose.

The duplicated text from Yu et al., 2016. Revised to Line 424-436

 The FT-IR spectrum of levan showed a strong, wide band between 3600 cm-1 and 3200 cm -1 corresponding to O-H stretching vibrations that exist in polymeric alcohol. Moreover, peaks around 3437, 2928 and 2885 cm -1 corresponding to O-H stretching vibrations and C-H stretching and bending vibrations indicated the existence of polysaccharide. The bending vibration of O-H associated with an intense peak around 1651 cm-1 also indicated the presence of bound water.

The stretching vibrations of C-O-C in pyranose or furanose were attributed to peaks around 1126 and 1014 cm-1 , while peaks around 922 and 806 cm-1 were associated with the symmetrical stretching vibration of furanose and the bending vibration of D-type C-H bonds present in furanose, respectively. Thus, typical signals from furanose were completely different from the signals of pyranose, enabling these two components to be easily distinguished. The existence of C-O-C signals was in accordance with this result. Therefore preliminary assessment of EPS structure showed that it was composed mostly from a D-furanose with no substituent groups.

The duplicated text from Gojgic-Cvijovic et al., 2018. Revised to Line 472-475

 Levans from the most-studied Gram positive bacterial genera, Bacillus and Paenibacillus, have molecular weights in the 104 to 109 Da range [71, 72], while levan from B. licheniformis 8-37-0-1 has a molecular weight of 2.826 x104 Da [73]. However, the levan produced by Kozakia baliensis has an exceptionally high molecular weight of 2.466 x 109 Da [70]. 

The duplicated text from Xu et al., 2016. Revised to Line 476-479

 The molecular weights of microbial levans are quite variable depending on the specific producer and the cultivation conditions [74]. For example B. subtilis (natto) produced both low (11 kDa) and high (1800 kDa) molecular weight levans. However, B. polymyxa (NRRLB-18475) produced only very high molecular weight levan of approximately 2 × 106 Da [71, 75].

The duplicated text from Gojgic-Cvijovic et al., 2018. Revised to Line 492-495

Indeed, similar effect was previously reported. Relatively high molecular weight levan (range > 106 Da) is produced in sugar beet molasses-based media with 200 g L-1 total sucrose, while relatively lower molecular weight levan (in the range of 105 Da) is produced in medium with high concentrations (400 g L-1) of sucrose [49]. 

The duplicated text from Revin et al., 2020. Revised to Line 500-509

Samples used for structural analysis were prepared by freeze-drying. The SEM showed that the EPS surface has complex petal morphology with rounded macropores (Fig 9). The structure of the sample was similar to that of an aerogel, although it is known that levan cannot be gelled without a surface modification. Aerogels are known as highly porous materials with a huge internal surface area [77]. Two methods usually applied to form aerogels are (1) supercritical drying or (2) freeze-drying. In both cases, the structure of the material is preserved during drying. 

An analysis of the sample composition was performed using energy-dispersive X-ray spectroscopy (EDX) and the elements present in the sample were analyzed by (S5 Fig). The results demonstrated strong C and O signals and allowed the identification of 57.49% carbon and 41.77% oxygen corresponding to groups present in the levan structure

The duplicated text from Revin et al., 2020. Revised to Line 512-519

The stability and longevity of polymers are usually measured by thermogravimetric analysis. One important characteristic of EPS is its thermostability, especially for high-temperature applications. Indeed, levan can be fully thermally processed through molding and extrusion methods [4]. Therefore, we conducted a thermogravimetric study of the EPSs produced by P. polymyxa 2020 during growth in sucrose and molasses media (Fig 10). The thermal weight loss of the EPS started at 57°C, and when the treatment reached 200 °C, the EPS lost about 15% of its mass. Thermogravimetric analysis demonstrated that levan begin to degrade at 200°C, and by the time the temperature reached 300 °C about 60% of the EPS mass has been lost.

The duplicated text from Revin et al., 2020. Revised to Line 522-532

 To investigate the effect of levan produced by P. polymyxa 2020 from a sucrose and molasses media, cytotoxicity studies were performed to test the proliferation of a mouse fibroblast cell culture L929. The viability of L929 cells was evaluated by the MTT assay (S4 Fig)

All the materials tested showed very low toxicity. No reduced cell viability was seen during incubation with levan produced by P. polymyxa 2020 either in a sucrose or a molasses media. Thus, EPS does not inhibit the proliferation of L929 cells. We also observed that the morphology of L929 cells was not affected by levan produced by P. polymyxa 2020 in a sucrose and molasses media. The cells were monitored using an inverted optical microscope and the morphologies of untreated L929 cells were similar to those that had been treated with EPS. These results indicate that levan produced by P. polymyxa 2020 is a promising source of functional biomaterials for a wide range of cellular applications, including biomedicine.

The duplicated text from Gojgic-Cvijovic et al., 2019. Revised to Line 45-49

It has received much attention due to its anticancer, anti-oxidant, antibacterial, anti-inflammation, immunomodulating and prebiotic activities as well as many outstanding physicochemical properties such as low intrinsic viscosity, high adhesive strength, high water solubility, film-forming ability and high biocompatibility [2–4].

Review Comments to the Author

Reviewer 1

This study reports about isolation of Paenibacillus polymyxa 2020 from wasp honeycombs capable of producing levan. Whole genome sequencing and methylome analysis of P. polymyxa 2020 were performed and bioinformatic analysis identified putative levan synthetic operon. sacC and sacB genes were further cloned and their products identified. To reduce the production cost inexpensive substrate molasses was investigated for EPS production. Manuscript needs following changes to improve it further.

Authors' response: Thank you very much for all of your detailed comments and suggestions. We found them quite useful as we approached our revision.

Comments:

1. Remove word novel from title as many Paenibacillus strains already isolated and reported for different applications.

Authors' response: Thank you very much for your recommendation. The word novel was removed from the title.

2. In introduction part add information related to different exopolysaccharides, microorganisms, and their applications. Also add information related to market potential of levan and other EPS.

Authors' response: Information related to different exopolysaccharides and microorganisms was added:

Some of the most used EPSs are xanthan from Xanthomonas campestris, dextran produced by Leuconostoc mesenteroides, Streptococcus sp., Lactobacillus sp., alginate from Azotobacter sp., Pseudomonas sp., gellanderived from Pseudomonas elodea and Sphingomonas paucimobilis, curdlan from Alcaligenes faecalis, hyaluronan from Streptococcus equi, pullulan from Aureobasidium pullulans, bacterial cellulose from Komagataeibacter sp., and levan from Bacillus sp., Paenibacillus sp., Halomonassp. etc.

Information related to the market potential of EPS was added:

The importance attached to the EPS market and commercial applications encourages the rapid development of research to source new producers, the isolation and production of EPS, and obtaining novel functional materials for a wide range of applications

3. Line 138: Why author decided to use 10% v/v inoculum? Why author didn’t try to optimize various culture conditions for optimum growth and EPS production, as this was a new isolate, and these parameters may help to increase production.

Authors' response: We plan to present data on the optimization of the various culture conditions for optimum growth and EPS production (the amount of inoculum, the mixing rate, the degree of aeration, etc.), including during the scaling process in bioreactors of various volumes, in the next article due to the large volume of this article.

4.What are other components present in EPS? As EPS is composed of carbohydrates, proteins, organic acids etc.

Authors' response: The Fourier transform infrared (FT-IR) and nuclear magnetic resonance (NMR) spectra demonstrate that the EPS is a linear β-(2→6)-linked fructan (levan). No other components were found in the EPS. This is also confirmed by the results of SEM-EDX analysis of samples (oxygen - 41.77%, carbon – 57, 49%) (Supporting Information S5Fig).

5.Line 339: P. polymyxa 2020 showed a high content of EPS in comparison to other strain at almost same concentration of sucrose. Compare in terms of culture conditions also (pH and T) used in this study and other study to provide better information to readers.

Authors' response: A comparison of culture conditions has been added.

6. Provide figure number to all the figures.

Authors' response: Figure number are provided for all the figures.

7. Remove grid lines from the figures

Authors' response: Grid lines are removed from the figures.

8. Line 99: C/N and pH plays important role in EPS production, while in this study no such experiment included. Need to perform experiment and include results.

Authors' response: We plan to present data on the optimization of the various culture conditions for optimum growth and EPS production (the amount of inoculum, the mixing rate, the degree of aeration, etc.), including during the scaling process in bioreactors of various volumes, in the next article due to the large volume of this article. Information about change in pH during growth was added (S1 Fig, S2 Fig).

Reviewer 2

Thank you very much for all of your detailed comments and suggestions. We found them quite useful as we approached our revision.

Comments:

The paper is interesting in describing a new strain with relevant biosynthetic pathways to exopoysaccharide production, and the ability to achieve high titer of EPS. The authors run a wide array of characterization experiments, from a biomolecular and biological side to a more chemical-structure analyses.

However the manuscript fails to correctly describe in detail the experimental part.

I think there is a need to improve the methodologies section few examples are following:

Improvement of fermentation descriptions, give detains on growth.

Authors' response: Information about growth and change in pH during growth was added (S1 Fig, S2 Fig).

explain how they can purify and isolate EPS with such a simple 2 step process (comparison to other EPS, contamination of polysaccharides from teh media...etc.)

Authors' response: The Fourier transform infrared (FT-IR) and nuclear magnetic resonance (NMR) spectra demonstrate that the EPS is a linear β-(2→6)-linked fructan (levan). No other components were found in the EPS. This is also confirmed by the results of SEM-EDX analysis of samples (oxygen - 41.77%, carbon – 57, 49%) (Supporting Information FigS5).

Clarify why they use high temperature drying

Authors' response: Samples used for structural analysis were prepared by freeze-drying.

improve SEM analyses description

Authors' response: SEM analyses description was improved.

compared their limitation in NMR to eventually previously reported spectroscopic analyses of heteropolysaccarides like hyaluronic acid, or other GAGs.;

Authors' response: There are more than 10,000 papers describing the NMR spectroscopic analyses of heteropolysaccharides, and most of them provides unique fingerprinting data. Due to this, comparison of spectra to those of other polysaccharides is beyond the scope of this study. The comparative analysis of these data lies behind the usage of automated matching and subsequent manual verification of spectroscopic observables, as described in the manuscript (lines 438 PONE-S-20-50060.pdf, p19 in the revised manuscript, Table 1). To clarify the NMR analysis we refactored the NMR section (lines 413-450 in PONE-S-20-50060.pdf, lines 414-467 in the revised manuscript).

Explain the meaning of the cytotoxicity study with respect to a proposed application;

Authors' response: These results showed that levan produced by P. polymyxa 2020 is a promising candidate for obtaining functional biomaterials for a wide range of applications, including biomedicine. Therefore, it is non-toxic for cell culture applications.

e) describe the specific interest in the molecular determinant of the biosynthetic pathways, since they report already a very high production (and productivity) when compared to other microbial polysaccharides.

Authors' response: We plan to study the biosynthetic pathway in more detail in the future.

Minor points are also highlighted below: The shake flasks growth, although a simple and routine approach should be better described here (biomass density achieved, growth curve, or any othe information like change in pH during growth.

Authors' response: Information about biomass and change in pH during growth was added (S1 Fig, S2 Fig).

If the process is a fully aerobic one it seems that the inoculums of 100 mL medium in a 250 mL flask is not ensuring a sound gas distribution, so is limiting oxygen transfer, this especially in relation to the increase n velocity during growth.

Authors' response: In our work, a fairly high mixing speed was used -250 rpm.

In Figure 2 the production of levan is presented without any information on the biomass growth. Since the authors are using a very high sugar content (either a disaccharide or a complex mixtures ..also containing oligo saccharides) the osmotic pressure will be different in the various condition and thus I will expect a difference in growth rate and then of Levan production.

Authors' response: Information about biomass growth was added (S1 Fig, S2 Fig).

The reference at time zero (for zero amount of levan) was quantified on the medium of this was an assumption of the authors? Please specify.

Authors' response: The concentration of levan was measured right after addition of inoculum

The downstream procedure described is very simple, I argue it is very difficult to separate pure EPS only using precipitation and dialysis, did the author apply the same procedure to the fermentation broth to analyze similar residues directly coming from the complex components (e.g. yeast extract or other) 

Authors' response: We applied the same procedure to the fermentation broth. However, no precipitation was observed. Repeated precipitation and dissolution in water was performed to purify the levan. The Fourier transform infrared (FT-IR) and nuclear magnetic resonance (NMR) spectra demonstrate that the EPS is a linear β-(2→6)-linked fructan (levan). No other components were found in the EPS. This is also confirmed by the results of SEM-EDX analysis of samples. The results showed strong signals of C and O (oxygen - 41.77%, carbon – 57, 49%) (Supporting Information S5 Fig).

Fig 10. SEM images of EPS of P. polymyxa 2020 at a various magnification- please improve description in the legend

Authors' response: SEM image and description in the legend were changed.

SEM micrograph should be better explained in the text (how the films were obtained, and with procedeure before metalization. It is not clear why a porosity should be found if not for very low molecular weight components (like salts) that may leave the film if washed. The authors should clarify this point

Authors' response: SEM analyses description was changed:

Samples used for structural analysis were prepared by freeze-drying. The SEM showed that the EPS surface has complex petal morphology with rounded macropores (Fig 9). The structure of the sample was similar to that of an aerogel, although it is known that levan cannot be gelled without a surface modification. Aerogels are known as highly porous materials with a huge internal surface area [77]. Two methods usually applied to form aerogels are (1) supercritical drying or (2) freeze-drying. In both cases, the structure of the material is preserved during drying. 

An analysis of the sample composition was performed using energy-dispersive X-ray spectroscopy (EDX) and the elements present in the sample were analyzed by (S5 Fig). The results demonstrated strong C and O signals and allowed the identification of 57.49% carbon and 41.77% oxygen corresponding to groups present in the levan structure

Figure 12 is very confusing, and low quality

Authors' response: Fig12 has been replaced with a higher quality Fig11, originally submitted gel image for panel C was lack of protein markers, therefore it was replaced with a new image. As a request of PlosOne we provide original unadjusted and uncropped images as supporting S7 Fig.

---

## [Decision Letter · Decision Letter 1]

20 May 2021

PONE-D-20-40095R1

Production and сharacterization of the exopolysaccharide from strain Paenibacillus polymyxa 2020

PLOS ONE

Dear Dr. Formenkov,

Thank you for submitting your manuscript to PLOS ONE. After careful consideration, we feel that it has merit but does not fully meet PLOS ONE’s publication criteria as it currently stands. Therefore, we invite you to submit a revised version of the manuscript that addresses the points raised during the review process.

Please see below for my specific comments.

We look forward to receiving your revised manuscript.

Kind regards,

Dharam Singh

Academic Editor

PLOS ONE

Additional Editor Comments (if provided):

Dear Authors,

Although, revision is much improved but still I feel that there are concerns that need to be corrected before making final decision. Please see the following concerns.

1. Writing part is poorly managed through the manuscript. Basically, each paragraph should be arranged as a unit that should be in coherent with another para. But, if you see text has been arranged as small small (5-6 lines) distributed text. See lines 65-75. They can be come under single para. Similar is the problem throughout the introduction, materials and methods section, results and discussion. To refer few of them as lines 353-394, 414-424, 478-491 etc..

2. Supplementary S6: correct the concentration from 0,31 to 0.31 mg/ml. Similarly for other values too.

3. S7 Fig. is not labeled. Position of Protein marker is not defined.

4. What is the PI of cloned proteins? May be the negative charge of proteins have led to slow migration in the gel in S7Fig.

5. Besides, there are still several mistakes through the manuscript that need to be corrected.

Reviewers' comments:

Reviewer's Responses to Questions

**Comments to the Author**

1. If the authors have adequately addressed your comments raised in a previous round of review and you feel that this manuscript is now acceptable for publication, you may indicate that here to bypass the “Comments to the Author” section, enter your conflict of interest statement in the “Confidential to Editor” section, and submit your "Accept" recommendation.

Reviewer #1: All comments have been addressed

2. Is the manuscript technically sound, and do the data support the conclusions?

Reviewer #1: Yes

3. Has the statistical analysis been performed appropriately and rigorously? 

Reviewer #1: Yes

4. Have the authors made all data underlying the findings in their manuscript fully available?

Reviewer #1: Yes

5. Is the manuscript presented in an intelligible fashion and written in standard English?

Reviewer #1: Yes

6. Review Comments to the Author

Reviewer #1: Author has revised the manuscript according to the reviewers comment. Manuscript is recommended for publication in its present form.

7. PLOS authors have the option to publish the peer review history of their article (what does this mean?). If published, this will include your full peer review and any attached files.

Reviewer #1: **Yes: **Shashi Kant Bhatia

---

## [Author Response · Author response to Decision Letter 1]

1 Jun 2021

1.Writing part is poorly managed through the manuscript. Basically, each paragraph should be arranged as a unit that should be in coherent with another para. But, if you see text has been arranged as small (5-6 lines) distributed text. See lines 65-75. They can be come under single para. Similar is the problem throughout the introduction, materials and methods section, results and discussion. To refer few of them as lines 353-394, 414-424, 478-491 etc..

Dr. Roberts, a native English speaker has gone through the manuscript carefully and we hope that it now conforms to PLoS ONE standards

2. Supplementary S6: correct the concentration from 0,31 to 0.31 mg/ml. Similarly for other values too.

S6 Fig was corrected

3. S7 Fig. is not labeled. Position of Protein marker is not defined.

 S7 Fig was corrected, position of Protein marker was added

4. What is the PI of cloned proteins? May be the negative charge of proteins have led to slow migration in the gel in S7Fig.

Estimated isoelectric point of SacB protein is 5.194 (-10.741 charge at pH7.0) and isoelectric point of SacC protein is 8.869 (7.714 charge at pH7.0) (https://web.expasy.org/compute_pi/).

5. Besides, there are still several mistakes through the manuscript that need to be corrected.

Please see response to comment1

---

## [Editor Report · Decision Letter 2]

7 Jun 2021

Production and сharacterization of the exopolysaccharide from strain Paenibacillus polymyxa 2020

PONE-D-20-40095R2

Dear Dr. Formenkov,

We’re pleased to inform you that your manuscript has been judged scientifically suitable for publication and will be formally accepted for publication once it meets all outstanding technical requirements.

Kind regards,

Dharam Singh

Academic Editor

PLOS ONE
---

## [Editor Report · Acceptance letter]

23 Jun 2021

PONE-D-20-40095R2 

Production and сharacterization of the exopolysaccharide from strain *Paenibacillus polymyxa 2020 *

*Dear Dr. Fomenkov:*

*I'm pleased to inform you that your manuscript has been deemed suitable for publication in PLOS ONE. Congratulations! Your manuscript is now with our production department. *

*If your institution or institutions have a press office, please let them know about your upcoming paper now to help maximize its impact. If they'll be preparing press materials, please inform our press team within the next 48 hours. Your manuscript will remain under strict press embargo until 2 pm Eastern Time on the date of publication. For more information please contact onepress@plos.org.*

*If we can help with anything else, please email us at plosone@plos.org. *

*Thank you for submitting your work to PLOS ONE and supporting open access. *

*Kind regards, *

*PLOS ONE Editorial Office Staff*

*on behalf of*

*Dr. Dharam Singh *

*Academic Editor*

*PLOS ONE*